# NOISE IS ALL YOU NEED: SOLVING LINEAR INVERSE PROBLEMS BY NOISE COMBINATION SAMPLING WITH DIFFUSION MODELS

## ABSTRACT

Pretrained diffusion models have demonstrated strong capabilities in zero-shot inverse problem solving by incorporating observation information into the generation process of the diffusion models. However, this presents an inherent dilemma: excessive integration can disrupt the generative process, while insufficient integration fails to emphasize the constraints imposed by the inverse problem. To address this, we propose *Noise Combination Sampling*, a novel method that synthesizes an optimal noise vector from a noise subspace to approximate the measurement score, replacing the noise term in the standard Denoising Diffusion Probabilistic Models process. This enables conditional information to be naturally embedded into the generation process without reliance on step-wise hyperparameter tuning. Our method can be applied to a wide range of inverse problem solvers, including image compression, and, particularly when the number of generation steps $T$ is small, achieves superior performance with negligible computational overhead, significantly improving robustness and stability.

## 1 INTRODUCTION

Diffusion models have emerged as a powerful class of generative models, achieving state-of-the-art results in high-fidelity image generation, audio synthesis, video modeling, and even language modeling (Ho et al., 2020; Rombach et al., 2022; Podell et al., 2024; Ho et al., 2022; Nie et al., 2025; Gat et al., 2024). Beyond their impressive generative capabilities, researchers have also recognized their strong zero-shot potential for a variety of related tasks, including inpainting, depth estimation, segmentation, and classification (Lugmayr et al., 2022; Tian et al., 2024; Li et al., 2023).

Notably, it has been shown that diffusion models, without any additional training, can effectively solve general linear inverse problems by injecting information during the stochastic denoising steps (Wang et al., 2023; Chung et al., 2023; Cardoso et al.; Dou & Song, 2024; He et al., 2024; Kim et al., 2025). These tasks include denoising, inpainting, and super-resolution. Such powerful plug-and-play, training-free approaches have been adopted in more complex settings, including nonlinear and deep learning-guided generation tasks such as style transfer and image editing (Yu et al., 2023; He et al., 2024; Shi et al., 2024; Ye et al., 2024). However, since the imposed guidance drives the trajectory away from the manifold of real data and disrupts the consistency of the generation process, existing methods suffer from sampling instability and rely on complex hyperparameter tuning and long sampling schedules to compensate for the limited influence of the guidance (Wang et al., 2023; Song et al., 2023; Yang et al., 2024; Zhang et al., 2025).

In this paper, we propose *Noise Combination Sampling* (NCS), a novel framework that approximates the measurement score $\nabla_{\boldsymbol{x}_t} \log p(\boldsymbol{y} \mid \boldsymbol{x}_t)$ using a linear combination of Gaussian noise vectors to replace the noise term in the Denoising Diffusion Probabilistic Models (DDPM) process. By adjusting the sampling trajectory through constructed noise vectors rather than gradients, NCS mitigates the instability inherent in existing sampling-based inverse problem solvers, eliminating the need for carefully-tuned hyperparameters and preserving the denoising behavior of the generative model. Remarkably, the mathematically optimal combination weights can be derived in closed form via the Cauchy–Schwarz inequality, requiring negligible additional computation.

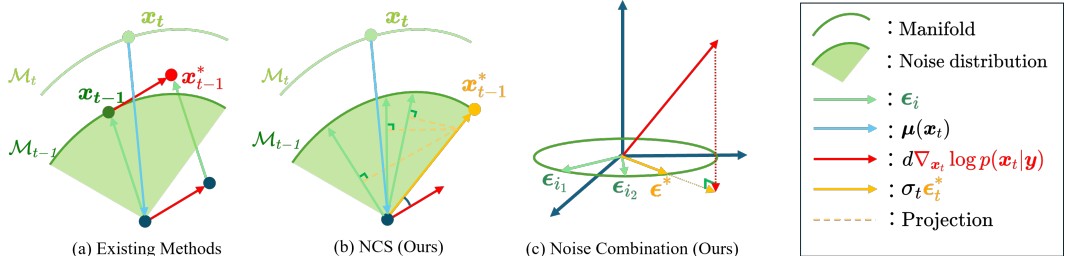

Figure 1: An illustration showing the difference between exact approximation methods and NCS. Existing methods cannot guarantee that the intervention—i.e., the measurement score—does not push the trajectory off the manifold $\mathcal{M}_{t-1}$ of $\boldsymbol{x}_{t-1}$. In contrast, NCS embeds the measurement score into the optimal noise within an ellipsoidal subspace, defined by the span of the noise codebook. This allows NCS to naturally preserve both the position of $\boldsymbol{x}_{t-1}$ on its manifold and the consistency of the diffusion process.

This approach can be seamlessly integrated into various mainstream inverse problem solvers, including Diffusion Posterior Sampling (DPS) (Chung et al., 2023) and Manifold-Preserving Gradient Descent (MPGD) (He et al., 2024), consistently yielding substantial improvements. Notably, a recently proposed impressive generative image compression method—Denoising Diffusion Codebook Models (DDCM) (Ohayon et al., 2025)—can be viewed as a special case of NCS, in which the top-$m$ noise vectors (with $m = 1$ by default) are selected from a codebook of size $K$, corresponding to an extremely low compression ratio. To improve compression quality, DDCM employs a greedy search with exponential complexity in $m$ to approximate the measurement score. In contrast, we demonstrate that a competitive quantization result can be achieved using the NCS solution with linear complexity.

Our contributions are summarized as follows:

- We propose the NCS framework, which leverages the noise variables in the DDPM process to approximate the measurement score in inverse problem solving. By optimally combining multiple Gaussian noise vectors, NCS synthesizes a noise sample that closely approximates the desired conditional distribution. We show that this optimization problem admits a closed-form solution via the Cauchy–Schwarz inequality.

- We demonstrate that prominent diffusion-based inverse problem solvers can be naturally reformulated under the NCS framework. Using DPS and MPGD as examples, we show the effectiveness of NCS. We further conjecture that most existing gradient-based approaches admit corresponding NCS formulations, offering a unified and principled perspective.

- We show that the generative image compression method DDCM can be viewed as a special case of NCS. By increasing the number of combined noise vectors and reducing the number of sampling steps, we significantly accelerate both compression and decompression, with negligible quality loss and linear complexity. This is possible because the complexity of the proposed quantization of the optimal noise combination is nearly independent of $m$.

- We conduct experiments across multiple datasets and inverse problem tasks, showing that NCS consistently outperforms existing methods that rely on careful hyperparameter tuning. In particular, NCS demonstrates remarkable robustness and stability when the number of diffusion steps $T$ is small, achieving high-quality results at reduced sampling cost.

## 2 BACKGROUND

### 2.1 DENOISING DIFFUSION PROBABILISTIC MODELS (DDPMS)

Diffusion models define the generative process as the reverse of a predefined noising process. Following the formulation of Song et al. (2021), we describe the forward (noising) process using an Itô stochastic differential equation (SDE), where $\boldsymbol{x}_t \in \mathbb{R}^d$ and $t \in [0, T]$:

$$d\boldsymbol{x}_t = f(\boldsymbol{x}_t, t) \, dt + g(t) \, d\boldsymbol{w}_t, \tag{1}$$

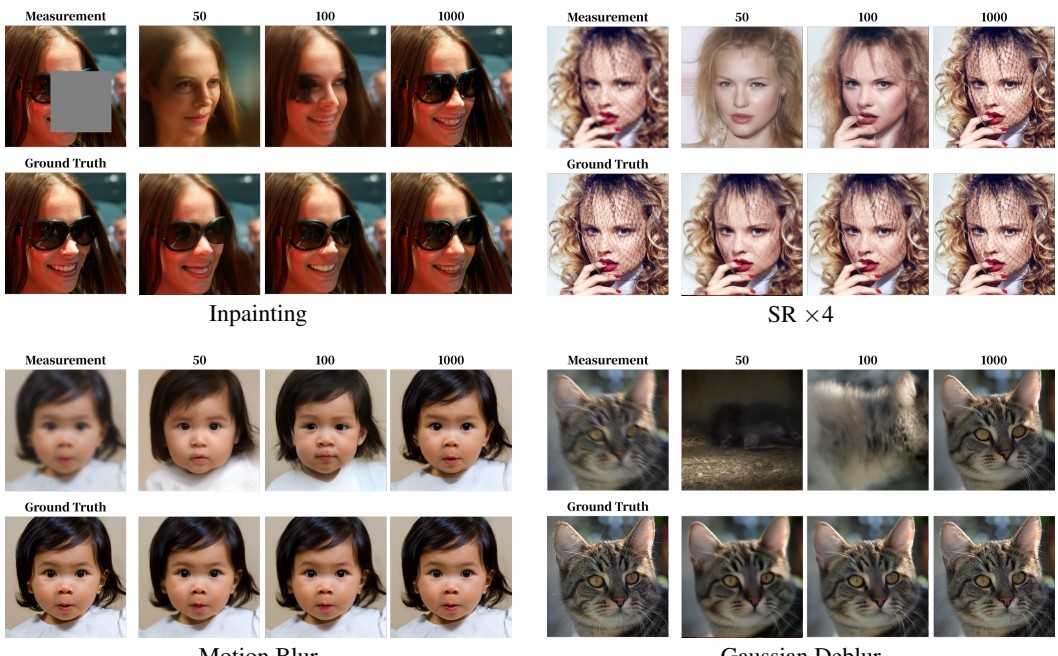

| Measurement | 50 | 100 | 1000 | Measurement | 50 | 100 | 1000 |

Inpainting                                        SR ×4

Motion Blur                                       Gaussian Deblur

Figure 2: Comparison of DPS and NCS-DPS across four inverse problems under varying sampling steps. NCS-DPS yields clearer details and greater stability, especially at small step counts.

where $f$ and $g$ denote the drift function and diffusion coefficient, respectively, and $\boldsymbol{w}_t$ is a standard Wiener process.

Assuming the initial data distribution is $\boldsymbol{x}_0 \sim p_{\text{data}}$ and the terminal distribution $\boldsymbol{x}_T$ is Gaussian, i.e., $\mathcal{N}(0, \mathbf{I})$, the goal of the generative (reverse) process is to recover samples from $p_{\text{data}}$ by reversing the diffusion trajectory. According to Anderson (1982), this can be achieved by solving the reverse-time SDE:

$$d\boldsymbol{x}_t = \left(f(\boldsymbol{x}_t, t) - g^2(t)\nabla_{\boldsymbol{x}_t} \log p_t(\boldsymbol{x}_t)\right) dt + g(t)\, d\boldsymbol{w}_t, \tag{2}$$

initialized at $\boldsymbol{x}_T \sim \mathcal{N}(0, \mathbf{I})$.

We follow Song et al. (2021)'s definition to choose a Variance-Preserving (VP)-SDE, or DDPM schedule to show the discrete update rule. Researchers usually use a neural network $\boldsymbol{s}_\theta(\boldsymbol{x}_t, t)$ to approximate the score function $\nabla_{\boldsymbol{x}_t} \log p_t(\boldsymbol{x}_t)$, which makes it possible to use the reverse process to generate the data. We consider the general condition by discretizing the whole process into T bins,

$$\boldsymbol{x}_{t-1} = \boldsymbol{x}_t - f(\boldsymbol{x}_t, t) + g^2(t)\boldsymbol{s}_\theta(\boldsymbol{x}_t, t) + g(t)\boldsymbol{\epsilon}_t \tag{3}$$

where $\boldsymbol{\epsilon}_t \sim \mathcal{N}(0, \mathbf{I})$.

Ho et al. (2020) consider the marginal distribution of $\boldsymbol{x}_t$ given $\boldsymbol{x}_0$ is Gaussian:

$$q(\boldsymbol{x}_t \mid \boldsymbol{x}_0) = \mathcal{N}\left(\boldsymbol{x}_t; \sqrt{\bar{\alpha}_t}\, \boldsymbol{x}_0,\ (1 - \bar{\alpha}_t)\, \mathbf{I}\right),$$

where the noise schedule is defined via $\beta_t = g(t) = -2f(t)$, $\alpha_t = 1 - \beta_t$, and $\bar{\alpha}_t = \prod_{s=1}^{t} \alpha_s$. This leads to the update rule of the DDPM process:

$$\boldsymbol{x}_{t-1} = \boldsymbol{\mu}(\boldsymbol{x}_t, t) + \sigma_t\, \boldsymbol{\epsilon}, \quad \boldsymbol{\epsilon} \sim \mathcal{N}(0, \mathbf{I}), \tag{4}$$

where $\boldsymbol{\mu}(\boldsymbol{x}_t, t) = \frac{1}{\sqrt{\alpha_t}}\left(\boldsymbol{x}_t - \frac{\beta_t}{\sqrt{1-\bar{\alpha}_t}}\, \boldsymbol{s}_\theta(\boldsymbol{x}_t, t)\right)$, and $\sigma_t = \sqrt{\beta_t}$ is the variance parameter governing the stochasticity of the reverse process.

The other important tool is Tweedie's formula in Kadkhodaie & Simoncelli (2021), which can be used to estimate the original signal $\boldsymbol{x}_0$ from a noisy observation $\boldsymbol{x}_t$ during the denoising process. In practice, DDPMs approximate this expectation using the trained score network, yielding:

$$\tilde{\boldsymbol{x}}_{0|t}(\boldsymbol{x}_t, t) = \mathbb{E}[\boldsymbol{x}_0 \mid \boldsymbol{x}_t] \approx \frac{1}{\sqrt{\bar{\alpha}_t}}\left(\boldsymbol{x}_t - \sqrt{1 - \bar{\alpha}_t}\, \boldsymbol{s}_\theta(\boldsymbol{x}_t, t)\right). \tag{5}$$

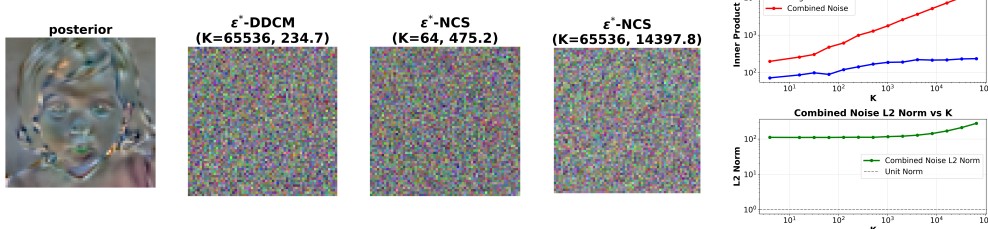

Figure 3: The comparison of the selected noise according to the original image. We choose a downsampled 64x64 image as the target image.

## 2.2 Linear Inverse Problems and Conditional Generation

Conditional generation addresses scenarios where only partial observations or measurements $\boldsymbol{y} \in \mathbb{R}^n$, derived from the original signal $\boldsymbol{x}_0 \in \mathbb{R}^d$, are available. The corresponding *inverse problem* is typically formulated as:

$$\boldsymbol{y} = \mathcal{A}(\boldsymbol{x}_0) + \boldsymbol{n}, \quad \boldsymbol{x}_0 \in \mathbb{R}^d, \ \boldsymbol{y}, \boldsymbol{n} \in \mathbb{R}^n, \tag{6}$$

where $\mathcal{A} : \mathbb{R}^d \to \mathbb{R}^n$ is a known linear degradation operator, and $\boldsymbol{n} \sim \mathcal{N}(0, \sigma^2 \mathbf{I})$ denotes additive white Gaussian noise. This formulation encompasses a wide range of tasks, including inpainting, super-resolution, and deblurring.

Solving the conditional generation problem with a pretrained diffusion model requires replacing the score function $\nabla_{\boldsymbol{x}_t} \log p_t(\boldsymbol{x}_t)$ in equation 2 with the conditional score function $\nabla_{\boldsymbol{x}_t} \log p_t(\boldsymbol{x}_t \mid \boldsymbol{y})$.

Under the Bayesian framework, the conditional distribution at $t$ is given by $p(\boldsymbol{x}_t \mid \boldsymbol{y}) = p(\boldsymbol{y} \mid \boldsymbol{x}_t)p(\boldsymbol{x}_t)/p(\boldsymbol{y})$, which indicates that $\nabla_{\boldsymbol{x}_t} \log p(\boldsymbol{x}_t \mid \boldsymbol{y}) = \nabla_{\boldsymbol{x}_t} \log p(\boldsymbol{y} \mid \boldsymbol{x}_t) + \nabla_{\boldsymbol{x}_t} \log p(\boldsymbol{x}_t)$. Substituting the $\nabla_{\boldsymbol{x}_t} \log p(\boldsymbol{x}_t \mid \boldsymbol{y})$ into the reverse SDE in equation 2 introduces an additional term that is not learned by the score network. The modified discrete update becomes:

$$\boldsymbol{x}_{t-1} = \boldsymbol{x}_t - f(\boldsymbol{x}_t, t) + g^2(t)\, \nabla_{\boldsymbol{x}_t} \log p_t(\boldsymbol{x}_t \mid \boldsymbol{y}) + g(t)\, \boldsymbol{\epsilon}_t \tag{7}$$

$$= \boldsymbol{x}_t - f(\boldsymbol{x}_t, t) + g^2(t)\, \nabla_{\boldsymbol{x}_t} \log p_t(\boldsymbol{x}_t) + g(t)\, \boldsymbol{\epsilon}_t + {\color{red} g^2(t)\, \nabla_{\boldsymbol{x}_t} \log p_t(\boldsymbol{y} \mid \boldsymbol{x}_t)}. \tag{8}$$

The red term in equation 7 captures the influence of the observation $\boldsymbol{y}$ on the sampling trajectory. Since the exact posterior $\nabla_{\boldsymbol{x}_t} \log p(\boldsymbol{y} \mid \boldsymbol{x}_t)$ is generally intractable, it typically requires an additional training process, such as classifier guidance or classifier-free guidance (Dhariwal & Nichol, 2021; Ho & Salimans, 2022) to be learned. The central challenge in diffusion-based inverse problem solving lies in accurately approximating $\nabla_{\boldsymbol{x}_t} \log p(\boldsymbol{y} \mid \boldsymbol{x}_t)$. Most methods, including explicit approximations and variational inference techniques Daras et al. (2024), directly modify the sampling trajectory during the denoising process, which can disrupt the consistency of the denoising process. In contrast, sampling-based methods (Dou & Song, 2024; Cardoso et al.; Wu et al., 2023) avoid altering the generative path by selecting candidate samples that satisfy the observation constraint through importance sampling or rejection. However, these approaches often suffer from weak guidance influence and high computational cost.

## 3 Noise Combination Sampling (NCS)

In the natural diffusion process, for all $t \in [0, T]$, $\boldsymbol{x}_t$ remains on its corresponding manifold, which is also the manifold on which the neural network $\boldsymbol{s}_\theta(\boldsymbol{x}_t, t)$ was trained. As illustrated in Figure 1, existing inverse problem solvers typically incorporate the measurement score $\nabla_{\boldsymbol{x}_t} \log p(\boldsymbol{y} \mid \boldsymbol{x}_t)$ as an external term, adding it directly during the denoising steps. This intervention disrupts the generative consistency of the diffusion process. In contrast, NCS does not alter the assumption or the method for approximating $\nabla_{\boldsymbol{x}_t} \log p(\boldsymbol{y} \mid \boldsymbol{x}_t)$. Instead, it implicitly embeds the conditional information into the noise component of the DDPM update, preserving the trajectory on the learned manifold. The modified update rule is given by:

$$\boldsymbol{x}_{t-1} \approx \boldsymbol{x}_t - f(\boldsymbol{x}_t, t) + g^2(t)\, \nabla_{\boldsymbol{x}_t} \log p_t(\boldsymbol{x}_t) + {\color{orange} g(t)\, \boldsymbol{\epsilon}_t^*}, \tag{9}$$

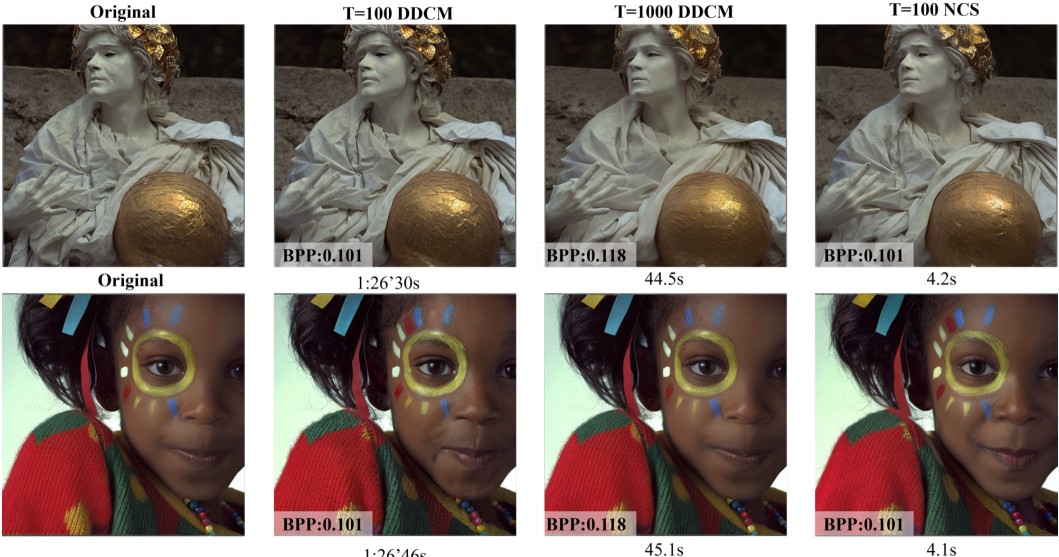

| Original | T=100 DDCM | T=1000 DDCM | T=100 NCS |
|---|---|---|---|

BPP:0.101    BPP:0.118    BPP:0.101

1:26'30s    44.5s    4.2s

BPP:0.101    BPP:0.118    BPP:0.101

1:26'46s    45.1s    4.1s

Figure 4: Comparison of the compression efficiency of NCS-MPGD and DDCM. For $T = 1000$, we choose $K = 32768$, $m = 12$, $C = 8$. For $T = 100$, we choose $K = 32768$, $m = 2$, $C = 0$. Our proposal get equivalent performance in fewer time.

where $\boldsymbol{\epsilon}_t^*$ is a constructed noise vector that approximates the effect of the conditional term. NCS seeks to find an optimal noise vector $\boldsymbol{\epsilon}_t^*$ within the span of a finite set of noise vectors, referred to as the noise codebook, and synthesizes it as a linear combination of these basis vectors to align with the target conditional direction. We formalize this process below.

**Theorem 1** (Noise Combination Sampling). *For linear inverse problems, the optimal noise vector $\boldsymbol{\epsilon}_t^*$ that best aligns with the conditional score direction is given by:*

$$\boldsymbol{\epsilon}_t^* = \sum_{i=1}^{K} \gamma_i \boldsymbol{\epsilon}_i, \tag{10}$$

*where $\{\boldsymbol{\epsilon}_i\}_{i=1}^{K}$ are standard Gaussian vectors from a fixed noise codebook, and $\boldsymbol{\gamma} = (\gamma_1, \ldots, \gamma_K)$ denotes the combination weights. The optimal weights are obtained by solving the following constrained optimization problem:*

$$\boldsymbol{\gamma}^* = \underset{\boldsymbol{\gamma} \in \mathbb{R}^K, \, \|\boldsymbol{\gamma}\|_2 = 1}{\operatorname{argmax}} \left\langle \nabla_{\boldsymbol{x}_t} \log p(\boldsymbol{y} \mid \boldsymbol{x}_t), \sum_{i=1}^{K} \gamma_i \boldsymbol{\epsilon}_i \right\rangle. \tag{11}$$

The synthesized noise vector $\boldsymbol{\epsilon}_t^*$ remains standard normal, i.e., $\boldsymbol{\epsilon}_t^* \sim \mathcal{N}(0, \mathbf{I})$, as a consequence of the unit-norm constraint $\|\boldsymbol{\gamma}\|_2 = 1$. A complete proof is provided in Appendix A.

Under the NCS framework, any approximated conditional score, or intervention, can be embedded into the noise term via an optimal linear combination. In the following, we demonstrate how several representative inverse problem solvers can be reformulated as special instances of NCS.

**Definition 1** (NCS-DPS). *Chung et al. (2023) approximates the conditional score using the gradient of a likelihood loss defined between the observation $\boldsymbol{y}$ and the estimated signal $\tilde{\boldsymbol{x}}_{0|t}$. Specifically,*

$$\nabla_{\boldsymbol{x}_t} \log p(\boldsymbol{y} \mid \boldsymbol{x}_t) = \nabla_{\boldsymbol{x}_t} \log \mathcal{N}(\boldsymbol{y}; \mathbf{A}\tilde{\boldsymbol{x}}_{0|t}, \sigma_t^2 \mathbf{I})$$

$$= \frac{1}{\sigma_t^2} \left( \frac{\partial \tilde{\boldsymbol{x}}_{0|t}}{\partial \boldsymbol{x}_t} \right)^\top \mathbf{A}^\top \left( \boldsymbol{y} - \mathbf{A}\tilde{\boldsymbol{x}}_{0|t} \right). \tag{12}$$

*Its NCS counterpart is obtained by aligning the synthesized noise vector with this gradient direction:*

$$\boldsymbol{\gamma}^* = \underset{\boldsymbol{\gamma} \in \mathbb{R}^K, \, \|\boldsymbol{\gamma}\|_2 = 1}{\operatorname{argmax}} \left\langle \left( \frac{\partial \tilde{\boldsymbol{x}}_{0|t}}{\partial \boldsymbol{x}_t} \right)^\top \mathbf{A}^\top \left( \boldsymbol{y} - \mathbf{A}\tilde{\boldsymbol{x}}_{0|t} \right), \sum_{i=1}^{K} \gamma_i \boldsymbol{\epsilon}_i \right\rangle. \tag{13}$$

**Definition 2** (NCS-MPGD). *He et al. (2024) proposes performing updates directly on the estimated signal $\tilde{x}_{0|t}$ rather than the latent variable $x_t$. By isolating the additional term introduced in its update rule, we obtain the following approximation of the measurement score:*

$$\nabla_{x_t} \log p(y \mid x_t) = -\lambda_t \sqrt{\bar{\alpha}_t} \nabla_{\tilde{x}_{0|t}} \|y - A\tilde{x}_{0|t}\|_2^2$$
$$= 2\lambda_t \sqrt{\bar{\alpha}_t} A^\top (y - A\tilde{x}_{0|t}), \tag{14}$$

*where $\lambda_t$ is a time-dependent step size. The corresponding NCS formulation is given by:*

$$\gamma^* = \underset{\gamma \in \mathbb{R}^K, \|\gamma\|_2=1}{\arg\max} \left\langle A^\top(y - A\tilde{x}_{0|t}), \sum_{i=1}^K \gamma_i \epsilon_i \right\rangle. \tag{15}$$

Notably, if $\tilde{x}_{0|t}$ is replaced with $x_t$, the formulation reduces to Score-Based Annealed Langevin Dynamics (ALD) Jalal et al. (2021). For a comprehensive analysis of the connections between these methods, refer to the survey in (Daras et al., 2024).

**Definition 3** (DDCM under the NCS Framework). *DDCM formulates compression as a special case of inverse problems, where $A = I$, $n = 0$, and $y = x_0$. The conditional score is approximated based on the difference between the ground-truth signal and the estimated reconstruction:*

$$\nabla_{x_t} \log p(y \mid x_t) \approx \frac{\sqrt{\bar{\alpha}_t}}{1 - \bar{\alpha}_t} (x_0 - \tilde{x}_{0|t}). \tag{16}$$

*DDCM selects one single noise vector from a predefined codebook via one-hot maximization:*

$$\epsilon_t^* = \underset{i \in \{1,\dots,K\}}{\arg\max} \left\langle x_0 - \tilde{x}_{0|t}, \ \epsilon_t^i \right\rangle, \tag{17}$$

*After the reconstruction, DDCM stores these index and can reconstruct the image with index and the generative model. Ohayon et al. (2025) extend this to general linear inverse problems by:*

$$\epsilon_t^* = \underset{i \in \{1,\dots,K\}}{\arg\max} \left\langle y - Ax_t, \ A\epsilon_t^i \right\rangle. \tag{18}$$

*Within the NCS framework, we replace $x_t$ with the estimated $\tilde{x}_{0|t}$ and extend the one-hot selection to a full linear combination. This yields the following optimization problem:*

$$\gamma^* = \underset{\gamma \in \mathbb{R}^K, \|\gamma\|_2=1}{\arg\max} \left\langle y - A\tilde{x}_{0|t}, \ A\sum_{i=1}^K \gamma_i \epsilon_i \right\rangle. \tag{19}$$

This is equivalent to NCS-MPGD, as both achieve their optimum when $\gamma^*$ aligns with the same direction. A formal proof is provided in Theorem 2 and Appendix B. While Ohayon et al. (2025) propose selecting the top-$m$ noise vectors from a codebook of size $K$, this can be viewed as a special top-$m$ case of NCS-MPGD with restricted support in $\gamma$.

In the following, we describe how to compute the optimal combination weights $\gamma^*$ that define the synthesized noise $\epsilon_t^*$ under the NCS formulation.

**Theorem 2** (Optimal Noise Combination). *Let $c = \nabla_{x_t} \log p(y \mid x_t)$ denote the approximated measurement score, and let $E_t = [\epsilon_t^1, \dots, \epsilon_t^K] \in \mathbb{R}^{n \times K}$ be the matrix formed by stacking $K$ standard Gaussian noise vectors. The optimal weight vector $\gamma^* \in \mathbb{R}^K$ that maximizes the inner product $\langle c, E_t \gamma \rangle$ subject to $\|\gamma\|_2 = 1$ is given by:*

$$\gamma^* = \frac{c^\top E_t}{\|c^\top E_t\|_2}. \tag{20}$$

*Proof.* Define $v = c^\top E_t \in \mathbb{R}^K$. Our goal is to solve:

$$\gamma^* = \underset{\|\gamma\|_2=1}{\arg\max} \langle v, \gamma \rangle.$$

By the Cauchy–Schwarz inequality,

$$\langle v, \gamma \rangle \leq \|v\|_2 \|\gamma\|_2 = \|v\|_2,$$

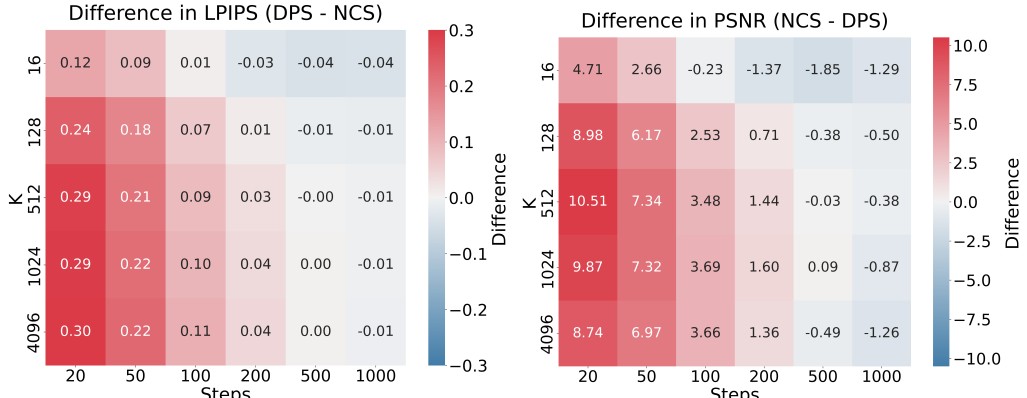

Figure 5: Influence of iteration and codebook size. We conduct the experiment on the inpainting task ($\sigma = 0$). The heatmap shows the improvement of the NCS-DPS compared to the DPS methods.

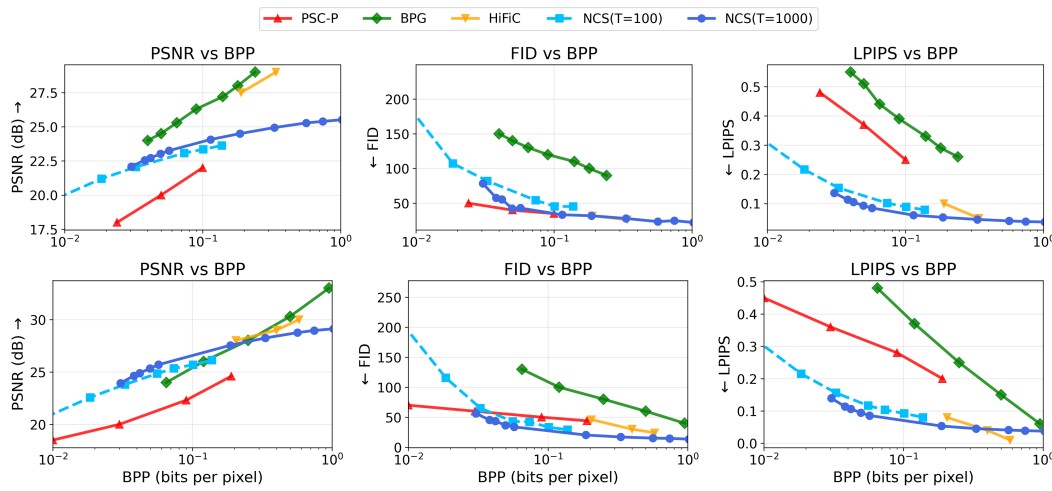

Figure 6: Comparison of compression methods. NCS achieves comparable reconstruction quality while reducing the number of compression steps from 1000 to 100, demonstrating significant efficiency with minimal quality loss.

with equality if and only if $\boldsymbol{\gamma}$ is aligned with $\boldsymbol{v}$, i.e., $\boldsymbol{\gamma} = \lambda \boldsymbol{v}$ for some scalar $\lambda$. Enforcing the constraint $\|\boldsymbol{\gamma}\|_2 = 1$ gives $|\lambda| = 1/\|\boldsymbol{v}\|_2$, and the maximum is attained at:

$$\boldsymbol{\gamma}^* = \frac{\boldsymbol{v}}{\|\boldsymbol{v}\|_2} = \frac{\boldsymbol{c}^\top \mathbf{E}_t}{\|\boldsymbol{c}^\top \mathbf{E}_t\|_2}.$$

$\square$

According to Theorem 2, the optimal noise combination can be directly computed based on inner products. This result also extends naturally to the top-$m$ case in DDCM-style problems. It significantly improves the efficiency of noise utilization: combining as few as 3 noise vectors can achieve an inner product magnitude comparable to that obtained by DDCM's selection from a codebook of 1024 noise vectors. Further details are provided in Fig. A and Appendix A.

### 3.1 COMPRESSION TASKS

DDCM observes that using more noise vectors with finer quantization bins can improve compression quality. To this end, it employs a greedy search strategy to identify the most suitable combination

of noise vectors across all quantization bins, selecting them iteratively (see Ohayon et al. (2025), Appendix B.5). However, this process results in an overall complexity that grows *exponentially* with the number of quantization bins, i.e., $2^C$, making it computationally expensive for large-scale settings. To address this limitation, we propose a method that directly approximates the quantization values using the closed-form solution from Theorem 2, reducing the computational cost to be negligible compared to the cost of noise inner product evaluations. Given the definition of bits per pixel (BPP) as $\text{BPP} = (T-1)\left(\log_2(K)m + C(m-1)\right)/n^2$, our formulation allows increasing both $m$ and $C$ for more accurate compression, while simultaneously reducing $T$. In Figures 4 and 6, we demonstrate that reducing $T$ by a factor of 10, while increasing both $m$ and the number of quantization bins, maintains the overall compression ratio, minimizes reconstruction quality loss, and yields a $10\times$ speedup in compression time. Further implementation details are provided in Appendix C.

## 3.2 APPLICATION SCOPE AND CHOICE OF CODEBOOK SIZE $K$

As shown in Fig. A, the NCS method approximates the measurement score by projecting it onto an ellipsoidal subspace spanned by $K$ noise vectors. As $K$ increases, the expressiveness of the noise combination improves, while the independence among the noise vectors diminishes. The norm of the synthesized noise grows on the order of $\sqrt{d}$, where $d$ denotes the data dimensionality. This behavior suggests that the synthesized noise may increasingly deviate from the standard noise distribution. In Fig. 5, the influence of $K$ on NCS performance appears quite robust. The NCS method performs well across a broad range of $K$ values, rendering it both practical and easy to deploy.

## 4 EXPERIMENTS

Table 1: Quantitative comparison of baseline solvers and their NCS variants on FFHQ dataset (Inpainting and SR 4×). Each cell shows PSNR (↑) / LPIPS (↓). Bold indicates the better result between a baseline and its NCS counterpart under the same setting.

| Task | Method | PSNR / LPIPS | | |
|---|---|---|---|---|
| | | 20 | 100 | 1000 |
| Inpainting (Box) | DPS | 12.52 / 0.497 | 18.67 / 0.286 | 22.71 / 0.139 |
| | **NCS-DPS** | **19.16 / 0.323** | **22.31 / 0.170** | **23.41 / 0.088** |
| | MPGD | 16.84 / **0.220** | 17.26 / 0.164 | 13.51 / 0.387 |
| | **NCS-MPGD** | **19.00** / 0.277 | **20.53 / 0.153** | **20.96 / 0.101** |
| | DAPS | 22.01 / 0.209 | **22.56** / 0.197 | 24.20 / **0.168** |
| | **NCS-DAPS** | **22.33 / 0.205** | 22.47 / **0.195** | **24.24** / 0.168 |
| Inpainting (Random) | DPS | 13.13 / 0.472 | 19.26 / 0.278 | 27.35 / 0.126 |
| | **NCS-DPS** | **21.20 / 0.297** | **27.31 / 0.137** | **31.57 / 0.042** |
| | MPGD | **21.80 / 0.172** | **25.45 / 0.100** | 25.05 / 0.214 |
| | **NCS-MPGD** | 20.25 / 0.290 | 25.21 / 0.126 | **28.71 / 0.049** |
| | DAPS | 14.08 / 0.625 | 16.31 / 0.543 | 25.33 / 0.238 |
| | **NCS-DAPS** | **16.832 / 0.557** | **19.50 / 0.456** | **25.73 / 0.230** |
| SR 4× | DPS | 12.87 / 0.480 | 16.70 / 0.338 | 23.53 / 0.171 |
| | **NCS-DPS** | **21.07 / 0.290** | **26.33 / 0.133** | **26.59 / 0.084** |
| | MPGD | 19.35 / 0.246 | 22.59 / 0.148 | 20.46 / 0.490 |
| | **NCS-MPGD** | **22.83 / 0.231** | **25.82 / 0.115** | **25.85 / 0.161** |
| | DAPS | **25.52** / 0.329 | **26.31 / 0.329** | **28.22 / 0.200** |
| | **NCS-DAPS** | 25.48 / **0.328** | 26.22 / 0.328 | 28.15 / 0.207 |

## 4.1 EXPERIMENTAL SETUP

All experiments were conducted on an NVIDIA RTX 4090 GPU. In Fig. 2, we choose pretrained models from Ho et al. (2020), and for experiments on FFHQ (Karras et al., 2019) and ImageNet (Deng et al., 2009) in Table 1 and 2, we choose models from Dhariwal & Nichol (2021). For compression experiments, we choose stable diffusion models 2.0 in Rombach et al. (2022).

## 4.2 INVERSE PROBLEM SOLVING

We evaluated our method on several challenging inverse problems: image inpainting, super-resolution, gaussian deblurring and motion deblurring. For inpainting, we tested both center box and random mask settings. For super-resolution, we examined 4× and 8×. Except for the experiments in Fig. 2, we choose DPS, MPGD, and DAPS Zhang et al. (2025) as the baseline solvers, inverse problems setting are the same as Chung et al. (2023), all tasks choose $\sigma = 0.05$. Detailed setting is shown in Appendix D. We use **black font** to highlight the better result between its baseline and NCS counterpart under the same setting. We use **blue bold font** to highlight the better result among all methods under the same setting. For other experiments, please refer to Appendix E.

## 4.3 COMPRESSION EXPERIMENTS

We conducted comprehensive experiments to evaluate the compression performance of our NCS-based approach. We compared our method against several state-of-the-art compression algorithms including PSC-P, BPG, and HiFiC (Albalawi et al., 2015; Elata et al., 2024; Muckley et al., 2023; Mentzer et al., 2020) on Kodak24 dataset (Franzen, 1999) and ImageNet dataset, using the same experimental setup as in DDCM Ohayon et al. (2025). The results demonstrate that our method maintains high fidelity while achieving significant storage savings compared to baseline approaches.

Table 2: Quantitative comparison of baseline solvers and their NCS variants on four inverse problems on ImageNet. Each cell shows PSNR (↑) / LPIPS (↓). Bold indicates the better result between a baseline and its NCS counterpart under the same setting.

| Task | Method | PSNR / LPIPS | | |
|------|--------|----|-----|------|
| | | 20 | 100 | 1000 |
| Inpainting (Box) | DPS | 11.67 / 0.682 | 15.34 / 0.550 | 18.52 / 0.308 |
| | **NCS-DPS** | **15.77 / 0.623** | **18.19 / 0.362** | **19.38 / 0.190** |
| | MPGD | 13.94 / 0.364 | 14.94 / 0.336 | 14.01 / 0.421 |
| | **NCS-MPGD** | **16.12 / 0.466** | **16.36 / 0.270** | **16.57 / 0.226** |
| | DAPS | **18.55** / 0.269 | **26.31 / 0.329** | **20.83** / 0.231 |
| | **NCS-DAPS** | 17.78 / **0.268** | 26.22 / 0.328 | 20.49 / 0.233 |
| Inpainting (Random) | DPS | 12.88 / 0.671 | 16.86 / 0.545 | 23.96 / 0.278 |
| | **NCS-DPS** | **17.66 / 0.596** | **23.70 / 0.300** | **28.69 / 0.097** |
| | MPGD | 17.15 / **0.285** | 19.32 / 0.435 | 16.38 / 0.770 |
| | **NCS-MPGD** | **17.47** / 0.495 | **22.05 / 0.222** | **24.02 / 0.178** |
| | DAPS | 25.59 / 0.252 | **26.31 / 0.329** | **28.79 / 0.146** |
| | **NCS-DAPS** | **25.97 / 0.240** | 25.75 / 0.318 | 28.64 / 0.149 |
| SR 4× | DPS | 12.12 / 0.688 | 14.92 / 0.587 | 20.40 / 0.365 |
| | **NCS-DPS** | **17.60 / 0.595** | **22.62 / 0.328** | **23.78 / 0.195** |
| | MPGD | 16.41 / 0.420 | 18.60 / 0.327 | 10.43 / 1.109 |
| | **NCS-MPGD** | **19.83 / 0.417** | **22.35 / 0.223** | **16.94 / 0.704** |
| | DAPS | **23.25 / 0.368** | **23.02 / 0.358** | **25.21 / 0.301** |
| | **NCS-DAPS** | 23.19 / 0.369 | 22.82 / 0.364 | 25.05 / 0.302 |

## 5 CONCLUSION

In this work, we introduced NCS, a principled framework that approximates the measurement score in diffusion models through an optimal linear combination of noise vectors. We derived a closed-form solution via the Cauchy–Schwarz inequality and showed that NCS unifies and generalizes existing inverse problem solvers, achieving strong performance with fewer diffusion steps and improved stability. With the NCS framework and its closed-form solution, we also accelerate DDCM while preserving similar reconstruction quality. As future work, we will explore applying NCS to compression, where its closed-form structure may inspire more efficient quantization schemes.

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

**Algorithm 1** Noise Combination Sampling for Linear Inverse Problems

---

**Require:** Codebooks $\mathcal{C}_t = \{\boldsymbol{\epsilon}_t^1, \ldots, \boldsymbol{\epsilon}_t^K\}$ for all $t$, represented as matrices $\mathbf{E}_t$; observation $\boldsymbol{y}$; approximate conditional score direction $\boldsymbol{c}$
**Ensure:** Reconstructed sample $\boldsymbol{x}_0$
 1: Sample initial latent $\boldsymbol{x}_T \sim \mathcal{N}(0, \mathbf{I})$
 2: **for** $t = T$ **to** $1$ **do**
 3:   $\tilde{\boldsymbol{x}}_{0|t} \leftarrow \text{Tweedie}(\boldsymbol{x}_t, t)$            (Eq. 5)
 4:   $\boldsymbol{c} \leftarrow \nabla_{\boldsymbol{x}_t} \log p(\boldsymbol{y} \mid \boldsymbol{x}_t)$
 5:   $\boldsymbol{\gamma}^* \leftarrow \boldsymbol{c}^\top \mathbf{E}_t / \|\boldsymbol{c}^\top \mathbf{E}_t\|_2$         (Eq. 20)
 6:   $\boldsymbol{\epsilon}_t^* \leftarrow \sum_{i=1}^K \gamma_i^* \boldsymbol{\epsilon}_t^i$
 7:   $\boldsymbol{x}_{t-1} \leftarrow \mu_\theta(\boldsymbol{x}_t, t) + \sigma_t \boldsymbol{\epsilon}_t^*$        (Eq. 4)
 8: **end for**
 9: **return** $\boldsymbol{x}_0$

---

## A   PROOF OF GAUSSIANITY AND OPTIMAL WEIGHTS

We first show that, under a unit-norm constraint on the combination weights, the synthesized noise remains standard normal, and then derive the closed-form optimizer for equation 11.

**Lemma 1** (Gaussianity of unit-norm combinations). *Let $\{\boldsymbol{\epsilon}_i\}_{i=1}^K$ be mutually independent with $\boldsymbol{\epsilon}_i \sim \mathcal{N}(\mathbf{0}, \mathbf{I})$. For any $\boldsymbol{\gamma} = (\gamma_1, \ldots, \gamma_K) \in \mathbb{R}^K$ with $\|\boldsymbol{\gamma}\|_2 = 1$ that is deterministic (or independent of $\{\boldsymbol{\epsilon}_i\}_{i=1}^K$), the linear combination*

$$\boldsymbol{\epsilon}_t^* = \sum_{i=1}^K \gamma_i \boldsymbol{\epsilon}_i$$

*satisfies $\boldsymbol{\epsilon}_t^* \sim \mathcal{N}(\mathbf{0}, \mathbf{I})$.*

*Proof.* By linearity of expectation, $\mathbb{E}[\boldsymbol{\epsilon}_t^*] = \sum_{i=1}^K \gamma_i \, \mathbb{E}[\boldsymbol{\epsilon}_i] = \mathbf{0}$. For the covariance, independence and isotropy give

$$\text{Cov}(\boldsymbol{\epsilon}_t^*) = \sum_{i=1}^K \sum_{j=1}^K \gamma_i \gamma_j \, \mathbb{E}[\boldsymbol{\epsilon}_i \boldsymbol{\epsilon}_j^\top] = \sum_{i=1}^K \gamma_i^2 \, \mathbf{I} = \|\boldsymbol{\gamma}\|_2^2 \, \mathbf{I} = \mathbf{I}.$$

Since $(\boldsymbol{\epsilon}_1, \ldots, \boldsymbol{\epsilon}_K)$ is jointly Gaussian and $\boldsymbol{\epsilon}_t^*$ is a linear transformation of it, $\boldsymbol{\epsilon}_t^*$ is Gaussian with mean $\mathbf{0}$ and covariance $\mathbf{I}$; hence $\boldsymbol{\epsilon}_t^* \sim \mathcal{N}(\mathbf{0}, \mathbf{I})$. $\qquad\square$

**Conclusion.** Combining Lemma 1 with Lemma 2, we obtain that the synthesized noise is standard normal whenever the synthesis codebook is independent of the weight computation, i.e., $\boldsymbol{\epsilon}_t^* \sim \mathcal{N}(\mathbf{0}, \mathbf{I})$.

In specific optimization scenarios, as the number of noise sources increases linearly, the inner product also increases nearly linearly according to $\log(K)$. This approach achieves significantly higher efficiency than selecting a single noise source within K (DDCM). Furthermore, the magnitude of the optimal noise remains constant over a considerable range (approximately equal to the average magnitude of the noise sources). Specific variations can be observed in Fig. A.

## B   PROOF OF EQUIVALENCE BETWEEN NCS OPTIMIZATION AND NCS-MPGD

**Theorem 3** (Equivalence of NCS Formulations). *For NCS optimization problem 11, we propose the following formulation: in which we replaced the $\boldsymbol{x}_0$ in Ohayon et al. (2025) with the more accurate reconstruction $\tilde{\boldsymbol{x}}_{0|t}$.*

$$\boldsymbol{\gamma}^* = \operatorname*{argmax}_{\boldsymbol{\gamma} \in \mathbb{R}^K, \|\boldsymbol{\gamma}\|_2 = 1} \left\langle \boldsymbol{y} - \mathbf{A}\tilde{\boldsymbol{x}}_{0|t}, \mathbf{A} \sum_{i=1}^K \gamma_i \boldsymbol{\epsilon}_i \right\rangle, \tag{21}$$

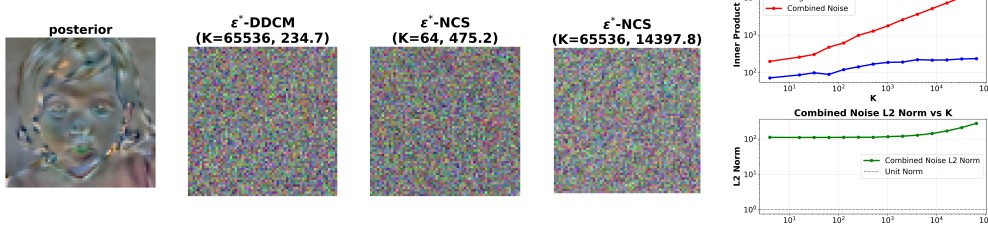

Figure 7: The comparison of the selected noise according to the original image. We choose a downsampled 64x64 image as the target image.

*It is equivalent to the NCS-MPGD formulation:*

$$\boldsymbol{\gamma}^* = \underset{\boldsymbol{\gamma} \in \mathbb{R}^K, \|\boldsymbol{\gamma}\|_2 = 1}{\operatorname{argmax}} \left\langle -\mathbf{A}^\top (\boldsymbol{y} - \mathbf{A}\tilde{\boldsymbol{x}}_{0|t}), \sum_{i=1}^K \gamma_i \boldsymbol{\epsilon}_i \right\rangle. \tag{22}$$

*That is, both optimization problems have the same optimal solution $\boldsymbol{\gamma}^*$.*

*Proof.* We prove this equivalence by showing that the two objective functions are identical for any feasible $\boldsymbol{\gamma}$.

Let $\boldsymbol{c} = \boldsymbol{y} - \mathbf{A}\tilde{\boldsymbol{x}}_{0|t}$ and $\boldsymbol{\varepsilon} = \sum_{i=1}^K \gamma_i \boldsymbol{\epsilon}_i$. The standard NCS objective can be written as:

$$\langle \boldsymbol{c}, \mathbf{A}\boldsymbol{\varepsilon} \rangle = \boldsymbol{c}^\top (\mathbf{A}\boldsymbol{\varepsilon}) \tag{23}$$
$$= (\boldsymbol{c}^\top \mathbf{A})\boldsymbol{\varepsilon} \tag{24}$$
$$= \boldsymbol{\varepsilon}^\top (\boldsymbol{c}^\top \mathbf{A})^\top \tag{25}$$
$$= \boldsymbol{\varepsilon}^\top \mathbf{A}^\top \boldsymbol{c} \tag{26}$$
$$= \left\langle \mathbf{A}^\top \boldsymbol{c}, \boldsymbol{\varepsilon} \right\rangle. \tag{27}$$

The NCS-MPGD objective is:

$$\left\langle -\mathbf{A}^\top \boldsymbol{c}, \boldsymbol{\varepsilon} \right\rangle = \left\langle -\mathbf{A}^\top \boldsymbol{c}, \sum_{i=1}^K \gamma_i \boldsymbol{\epsilon}_i \right\rangle \tag{28}$$

$$= \sum_{i=1}^K \gamma_i \left\langle -\mathbf{A}^\top \boldsymbol{c}, \boldsymbol{\epsilon}_i \right\rangle. \tag{29}$$

Since the two inner products are identical for every $\boldsymbol{\varepsilon}$ (and thus for every $\boldsymbol{\gamma}$), and the feasible set $\{\boldsymbol{\gamma} \in \mathbb{R}^K \mid \|\boldsymbol{\gamma}\|_2 = 1\}$ is the same for both problems, their maxima and argmax sets are identical.

Furthermore, since each objective is a linear functional of $\boldsymbol{\gamma}$, the maximum over the unit sphere occurs when $\boldsymbol{\gamma}$ is aligned with the functional's direction. For the standard NCS formulation, this gives:

$$\boldsymbol{\gamma}^* = \frac{(\mathbf{A}^\top \boldsymbol{c})^\top \mathbf{E}_t}{\|(\mathbf{A}^\top \boldsymbol{c})^\top \mathbf{E}_t\|_2}, \tag{30}$$

where $\mathbf{E}_t = [\boldsymbol{\epsilon}_1, \ldots, \boldsymbol{\epsilon}_K]$ is the matrix of noise vectors.

For the NCS-MPGD formulation, the optimal solution is:

$$\boldsymbol{\gamma}^* = \frac{(-\mathbf{A}^\top \boldsymbol{c})^\top \mathbf{E}_t}{\|(-\mathbf{A}^\top \boldsymbol{c})^\top \mathbf{E}_t\|_2} = \frac{-(\mathbf{A}^\top \boldsymbol{c})^\top \mathbf{E}_t}{\|(\mathbf{A}^\top \boldsymbol{c})^\top \mathbf{E}_t\|_2}. \tag{31}$$

The negative sign in the NCS-MPGD formulation is due to the maximization of the negative inner product, which is equivalent to minimizing the positive inner product. However, since we are maximizing the absolute value of the alignment, both formulations yield the same optimal direction (up to a sign, which is normalized out by the unit norm constraint).

Therefore, the two optimization problems are not merely equivalent—they are literally the same problem written in two different notations, and they achieve their maximum when the optimal $\gamma^*$ is the same (up to normalization). □

## C   QUANTIZATION BY NCS CLOSED-FORM SOLUTION

To utilize extra noise vectors to approximate the measurement score, Ohayon et al. (2025) proposes to use greedy search to find the noise vector that maximize the inner product between the measurement score and the noise vector, and to search the optimal quantization parameters on the selected noise vector and the next one. This process is computationally expensive. It will cost hours to search the optimal quantization parameters. if one chooses noise a noise quantization combination of $2^{10}$ with around 10 noise vectors, it will cost around 10 hours to search the optimal quantization parameters.

**Problem.**   Let $\{\epsilon_i\}_{i=1}^N$ be given noise vectors and let $c$ be the target. Define $b_i := \langle \epsilon_i, c \rangle$ and (optionally) align signs so $b_i \geq 0$ by replacing $\epsilon_i \leftarrow \mathrm{sgn}(b_i)\epsilon_i$. We form a mixture

$$\boldsymbol{m}(\boldsymbol{\gamma}) \;=\; \sum_{i=1}^m \gamma_i\, \boldsymbol{\epsilon}_{(i)}, \qquad \boldsymbol{\gamma} = (\gamma_1, \ldots, \gamma_m)^\top, \quad \sum_{i=1}^m \gamma_i^2 = 1,$$

where $(i)$ indexes an ordered subset of $m$ atoms (e.g., the Top-$m$ by $|b_i|$). The objective is to maximize alignment, i.e.

$$\max_{\boldsymbol{\gamma}} \; \langle \boldsymbol{m}(\boldsymbol{\gamma}), c \rangle \;=\; \max_{\boldsymbol{\gamma}} \; \sum_{i=1}^m \gamma_i\, b_{(i)} \quad \text{s.t.} \quad \sum_{i=1}^m \gamma_i^2 = 1.$$

**Closed-form (continuous) solution.**   Without quantization, the optimal coefficients on a fixed support are

$$\boldsymbol{\gamma}^* \;=\; \frac{\boldsymbol{b}_S}{\|\boldsymbol{b}_S\|_2}, \qquad \boldsymbol{b}_S = \big(b_{(1)}, \ldots, b_{(m)}\big)^\top,$$

by Cauchy–Schwarz. If the support $S$ is free, it is optimal to take the $m$ indices with largest $|b_i|$.

**Definition (quantization via L2 stick-breaking).**   We quantize by parameterizing $\boldsymbol{\gamma}$ through a stick-breaking map using

$$u_i \in \mathcal{Q} \subset (0, 1], \qquad i = 1, \ldots, 2^C - 1,$$

where $\mathcal{Q}$ is a finite grid with $2^{C-1} - 1$ elements (e.g., if m = 3, then if selected (not 0), $\mathcal{Q} = \{0.33, 0.66, 1\}$). The coefficients are then

$$\gamma_1 = \sqrt{u_1}, \tag{32}$$

$$\gamma_i = \Big( \prod_{j=1}^{i-1} \sqrt{1 - u_j} \Big) \sqrt{u_i}, \quad i = 2, \ldots, m - 1, \tag{33}$$

$$\gamma_m = \prod_{j=1}^{m-1} \sqrt{1 - u_j}. \tag{34}$$

By construction $\sum_{i=1}^m \gamma_i^2 = 1$ for any choices of $\{u_i\}$, so no final normalization is required. The inverse map (from any feasible $\gamma$ with $\sum \gamma_i^2 = 1$) is

$$u_1 = \gamma_1^2, \tag{35}$$

$$u_i = \frac{\gamma_i^2}{1 - \sum_{t=1}^{i-1} \gamma_t^2}, \quad i = 2, \ldots, m - 1. \tag{36}$$

**Using the closed-form to obtain a quantized solution.** We obtain a quantized solution directly from $\gamma^*$ in two simple steps:

1. **Project $\gamma^*$ into stick-breaking space.** Compute $\{u_i^*\}_{i=1}^{m-1}$ from $\gamma^*$ using the inverse map above. (When $b_{(1)} \geq b_{(2)} \geq \cdots$, $\gamma_i^* \propto b_{(i)}$ is non-increasing, which matches the stick-breaking order.)

2. **Quantize and reconstruct.** Independently quantize each stage by nearest-neighbor projection onto the grid,

$$\hat{u}_i \;=\; \arg\min_{u \in \mathcal{Q}} \big|u - u_i^*\big|, \qquad i = 1, \ldots, m-1,$$

then form $\hat{\gamma}$ from $\{\hat{u}_i\}$ via the forward stick-breaking map.

This yields $\hat{\gamma}$ in $O(m)$ time and preserves $\sum_i \hat{\gamma}_i^2 = 1$ by construction.

**Remark (stage-wise closed form and exact discrete refinement).** If one optimizes stage-wise in the *continuous* domain, the optimal fraction at stage $i$ has the closed form

$$u_i^\star \;=\; \frac{b_{(i)}^2}{b_{(i)}^2 + v_{i+1}^2}, \qquad v_m = b_{(m)}, \quad v_i = b_{(i)}\sqrt{u_i^\star} + v_{i+1}\sqrt{1 - u_i^\star}.$$

A discretized variant replaces $u_i^\star$ by the nearest grid point in $\mathcal{Q}$ at each stage (still $O(m)$). For the *exact* discrete optimum on $\mathcal{Q}$ one can use a 1D dynamic program:

$$v_m = b_{(m)}, \qquad v_i = \max_{u \in \mathcal{Q}} \Big\{ b_{(i)}\sqrt{u} + v_{i+1}\sqrt{1-u} \Big\},$$

which selects $\hat{u}_i \in \mathcal{Q}$ per stage and remains $O\big(m\,|\mathcal{Q}|\big)$.

## D    INVERSE PROBLEMS SETTING

We choose the same setting of inverse problems with Chung et al. (2023). Detailed setting is shown in Table 3 and 4.

As for the NCS method, because the strength of the measurement would influence the dimension of the noise space that is available for us to approximate, we set the $K$ for different tasks as follows:

- Super-resolution $\times 4$ (ImageNet/FFHQ): $K = 512$
- Super-resolution $\times 8$ (ImageNet/FFHQ): $K = 64$
- Inpainting box (ImageNet/FFHQ): $K = 64$
- Inpainting random (ImageNet/FFHQ): $K = 512$
- Gaussian deblurring (ImageNet/FFHQ): $K = 256$
- Motion deblurring (ImageNet/FFHQ): $K = 512$
- Phase retrieval (ImageNet/FFHQ): $K = 512$

## E    ADDITIONAL EXPERIMENTS OUTCOMES

We record experiments on Super-resolution $\times 8$, deblurring, gaussian blurring and phase retrieval here. The phase retrieval is too hard as an nonlinear inverse problem and evaluation would be vary unstable.

## F    COMPRESSION BY NCS-DPS

Since we have unified various methods under the NCS framework, this implies that—aside from DDCM, which is a special case equivalent to NCS-MPGD, NCS-DPS can also be employed for compression tasks. To evaluate this, we conducted experiments on several images from the Kodak24

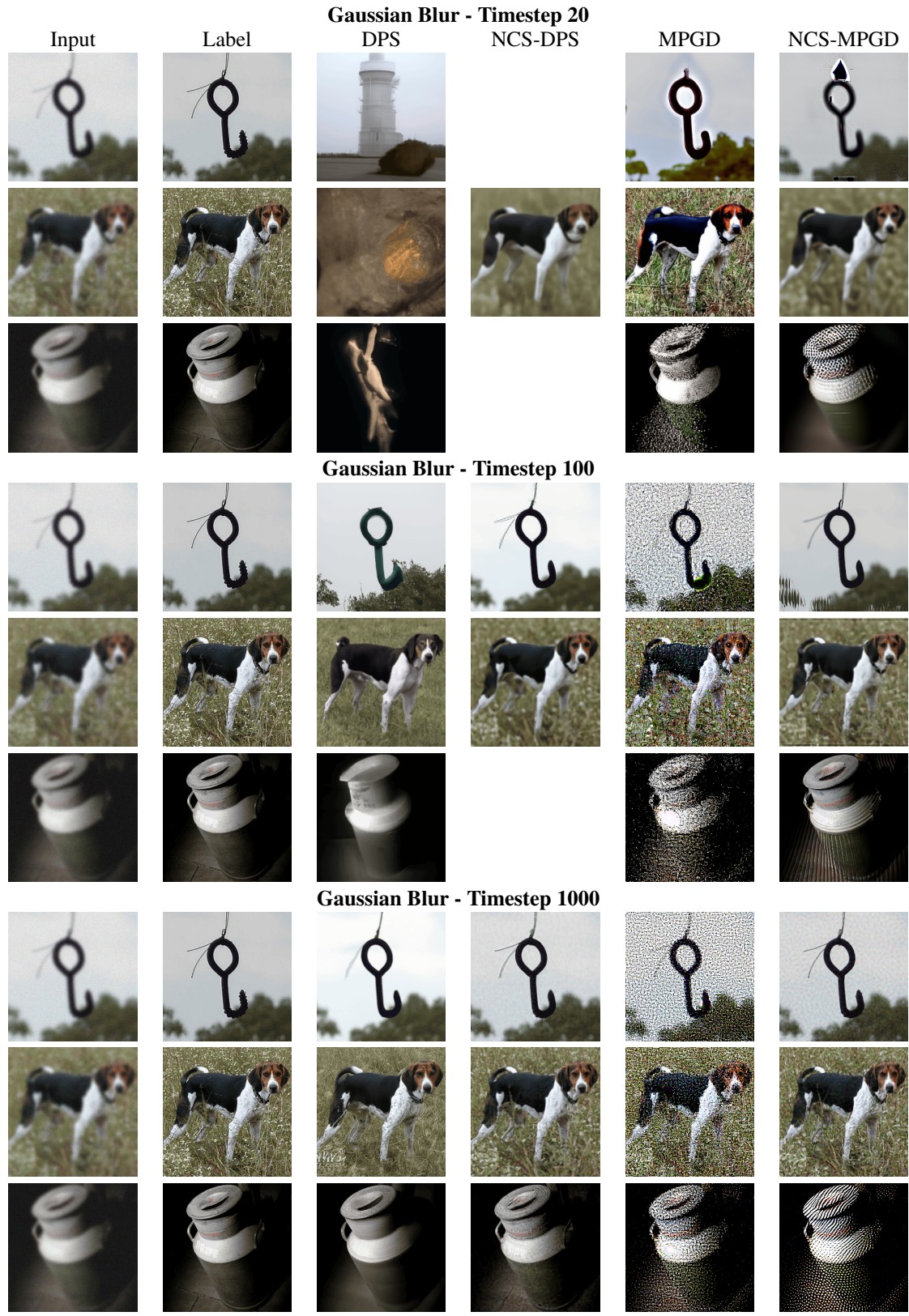

Figure 8: Visual comparison on Gaussian Blur task. Each row shows results for one image across different methods. The three sections correspond to timesteps 20, 100, and 1000 respectively.

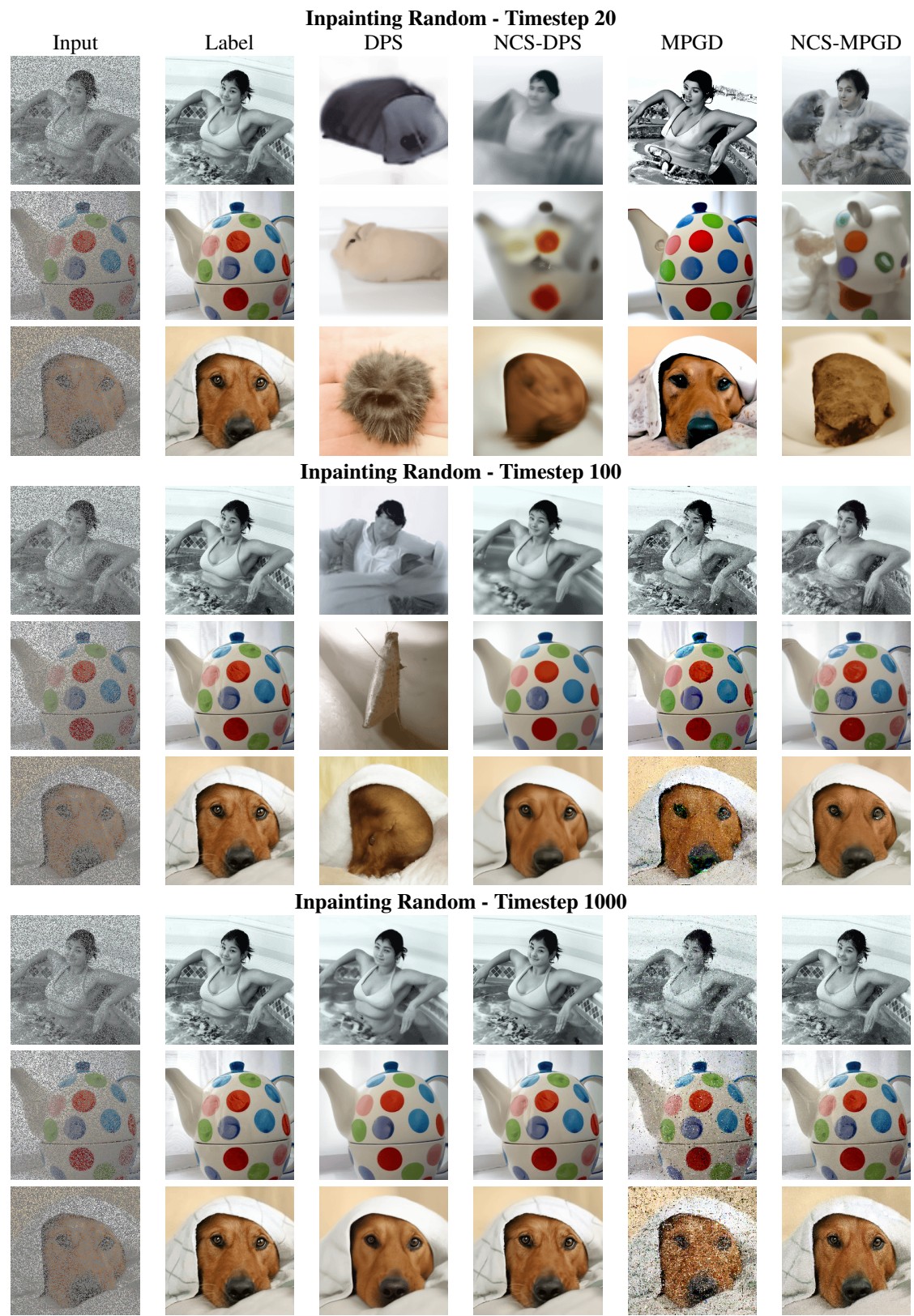

Figure 9: Visual comparison on Inpainting Random task. Each row shows results for one image across different methods. The three sections correspond to timesteps 20, 100, and 1000 respectively.

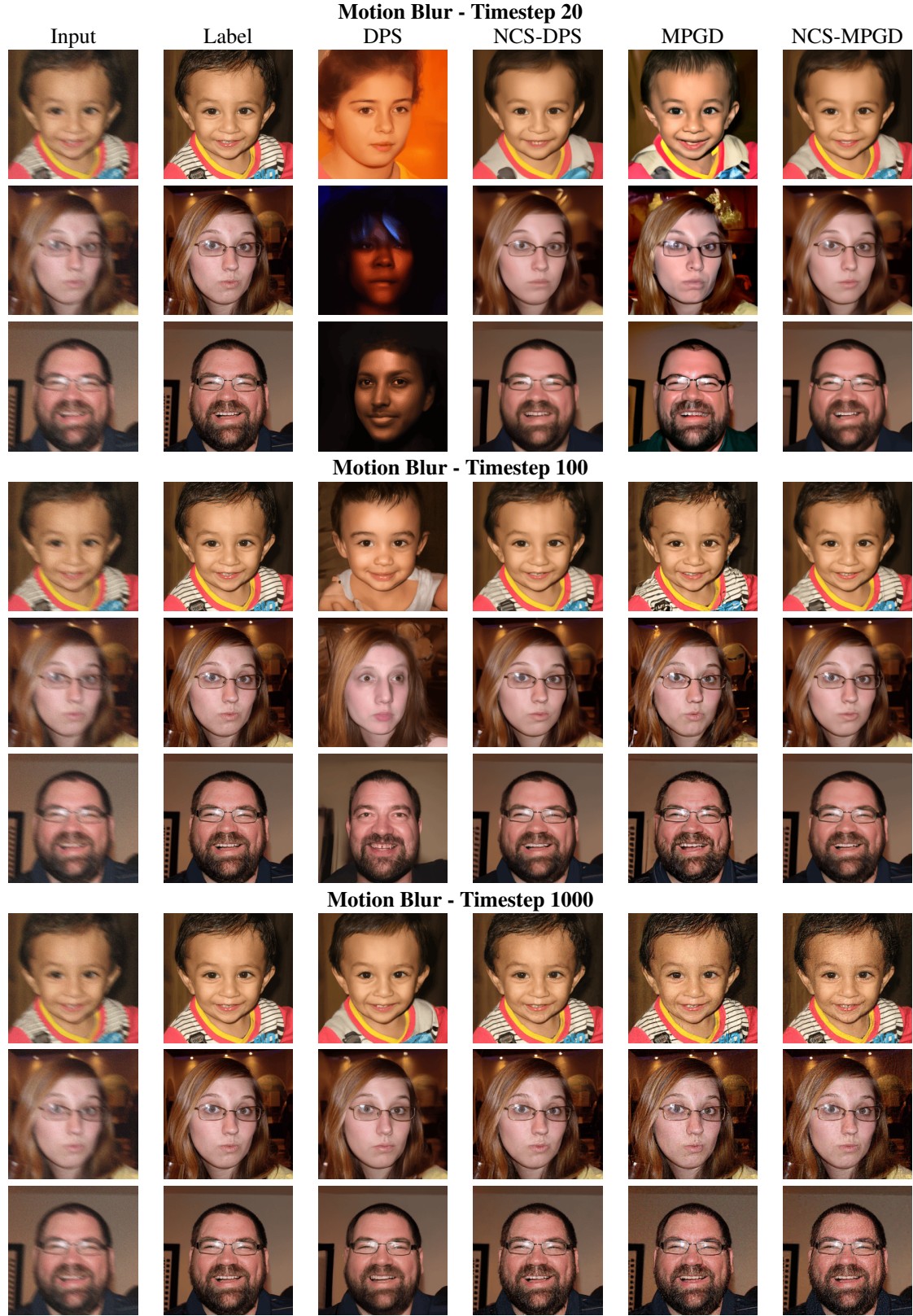

Figure 10: Visual comparison on Motion Blur task. Each row shows results for one image across different methods. The three sections correspond to timesteps 20, 100, and 1000 respectively.

**Super Resolution x4 - Timestep 20**

Input    Label    DPS    NCS-DPS    MPGD    NCS-MPGD

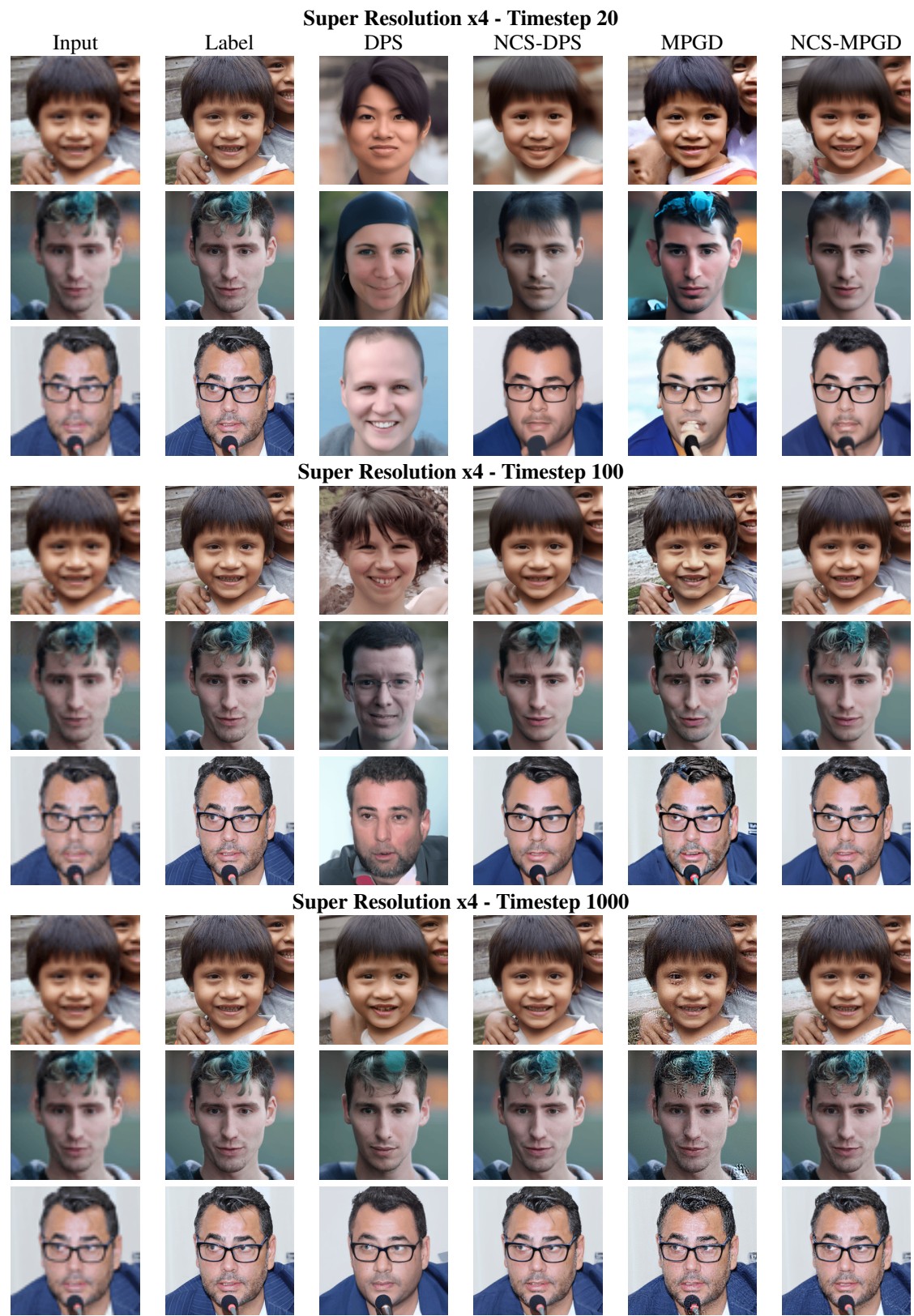

**Super Resolution x4 - Timestep 100**

**Super Resolution x4 - Timestep 1000**

Figure 11: Visual comparison on Super Resolution x4 task. Each row shows results for one image across different methods. The three sections correspond to timesteps 20, 100, and 1000 respectively.

Table 3: Quantitative comparison of baseline solvers and their NCS variants on the FFHQ dataset (Inpainting and SR 4×). Each cell shows PSNR (↑) / LPIPS (↓). Bold indicates the better result between a baseline and its NCS counterpart under the same setting.

| Task | Method | PSNR / LPIPS | | |
|---|---|---|---|---|
| | | 20 | 100 | 1000 |
| SR 8× | DPS | 11.84 / 0.5060 | 15.05 / 0.3869 | 20.87 / **0.2198** |
| | **NCS-DPS** | **20.83 / 0.2896** | **24.09 / 0.1535** | **21.97** / 0.2664 |
| | MPGD | 17.60 / 0.3032 | 20.08 / 0.2131 | 18.08 / 0.5603 |
| | **NCS-MPGD** | **22.10 / 0.2461** | **23.29 / 0.1703** | **21.67 / 0.3520** |
| | DAPS | 23.66 / 0.387 | 24.02 / 0.337 | 25.20 / **0.278** |
| | **NCS-DAPS** | **23.74 / 0.382** | **24.14 / 0.334** | **25.23** / 0.279 |
| Gaussian Deblur | DPS | 12.12 / 0.4948 | 20.02 / 0.2529 | 24.81 / 0.1298 |
| | **NCS-DPS** | **21.59 / 0.3111** | **26.68 / 0.1152** | **27.02 / 0.0825** |
| | MPGD | 21.36 / 0.2062 | 23.85 / 0.1336 | 24.80 / 0.1306 |
| | **NCS-MPGD** | **25.56 / 0.1954** | **26.83 / 0.1052** | **28.38 / 0.1292** |
| | DAPS | **25.80** / 0.319 | **26.63 / 0.267** | **28.38 / 0.187** |
| | **NCS-DAPS** | 25.80 / **0.318** | 26.57 / 0.268 | 28.34 / 0.189 |
| Motion Deblur | DPS | 12.21 / 0.4934 | 20.19 / 0.2514 | 25.87 / 0.1178 |
| | **NCS-DPS** | **22.05 / 0.3036** | **28.28 / 0.0974** | **29.50 / 0.0444** |
| | MPGD | 21.86 / 0.1914 | 24.88 / 0.1070 | 25.54 / 0.1416 |
| | **NCS-MPGD** | **26.48 / 0.1755** | **29.11 / 0.0752** | **27.08 / 0.1370** |
| | DAPS | **26.25 / 0.289** | **27.77 / 0.232** | **30.52** / 0.139 |
| | **NCS-DAPS** | 26.19 / 0.290 | 27.68 / 0.233 | 30.51 / **0.139** |
| Phase Retrieval | DPS | 8.15 / 0.6313 | 8.15 / 0.6313 | 10.75 / **0.5560** |
| | **NCS-DPS** | **12.95 / 0.4769** | **12.95 / 0.4769** | **12.06** / 0.5811 |
| | MPGD | **10.48 / 0.6347** | **10.48 / 0.6347** | 7.92 / 0.8084 |
| | **NCS-MPGD** | 6.99 / 0.7146 | 6.99 / 0.7146 | **8.75 / 0.7124** |

dataset, using the Stable Diffusion 2 model as the pre-trained backbone (Rombach et al., 2022). We adopted a codebook of size 1024 but restricted the selection to a single optimal noise vector to maintain consistency with the experimental settings of the DDCM paper.

As shown in Fig. 12, the compression efficiency of NCS-DPS is inferior to that of DDCM. The compressed images appear overly smooth, exhibiting a significant loss of fine details. This degradation may stem from inaccuracies in the gradient used during the reverse computation. Nonetheless, when using fewer denoising steps, the images compressed by NCS-DPS appear to retain richer semantic structures.

Table 4: Quantitative comparison of baseline solvers and their NCS variants on four inverse problems on ImageNet. Each cell shows PSNR (↑) / LPIPS (↓). Bold indicates the better result between a baseline and its NCS counterpart under the same setting.

| Task | Method | PSNR / LPIPS 20 | 100 | 1000 |
|------|--------|-------|------|------|
| SR 8× | DPS | 11.16 / 0.7016 | 13.78 / 0.6159 | 18.45 / **0.4150** |
| | **NCS-DPS** | **17.54 / 0.5833** | **21.02 / 0.3281** | **18.79** / 0.4529 |
| | MPGD | 15.46 / 0.4997 | 16.98 / 0.3833 | 10.86 / 1.0368 |
| | **NCS-MPGD** | **19.26 / 0.4612** | **19.74 / 0.3711** | **14.64 / 0.7984** |
| | DAPS | 21.43 / **0.368** | **21.32** / 0.472 | **25.21** / 0.301 |
| | **NCS-DAPS** | **21.50** / 0.369 | 21.24 / 0.472 | - / - |
| Gaussian Deblur | DPS | 12.02 / 0.6753 | 17.32 / 0.4932 | 21.01 / 0.2981 |
| | **NCS-DPS** | **13.18 / 0.6563** | **22.29 / 0.2984** | **23.87 / 0.2150** |
| | MPGD | 16.53 / 0.4127 | 14.72 / **0.2532** | 10.69 / 1.0786 |
| | **NCS-MPGD** | **20.40 / 0.4704** | **19.84** / 0.3627 | **20.31 / 0.4061** |
| | DAPS | **23.42 / 0.370** | **26.62** / 0.267 | **24.89** / 0.275 |
| | **NCS-DAPS** | 23.38 / 0.371 | 26.57 / 0.268 | 24.85 / 0.281 |
| Motion Deblur | DPS | 12.05 / 0.6757 | 17.39 / 0.5023 | 22.75 / 0.2679 |
| | **NCS-DPS** | **13.39 / 0.6482** | **24.13 / 0.2545** | **27.25 / 0.1027** |
| | MPGD | 16.99 / 0.3749 | 14.81 / 0.2525 | 13.99 / 0.8863 |
| | **NCS-MPGD** | **21.07 / 0.4419** | **24.49 / 0.2388** | **25.63 / 0.1933** |
| | DAPS | **24.54 / 0.297** | - / - | **27.93** / 0.181 |
| | **NCS-DAPS** | 24.42 / 0.298 | - / - | 27.85 / 0.185 |
| Phase Retrieval | DPS | **9.96 / 0.7513** | 7.77 / **0.7513** | 9.56 / 0.6875 |
| | **NCS-DPS** | 8.00 / 0.7350 | **8.99** / 0.8140 | **14.59 / 0.5693** |
| | MPGD | **9.26** / 0.8286 | **9.26** / 0.8286 | 7.62 / 0.8971 |
| | **NCS-MPGD** | 7.42 / **0.7545** | 7.42 / **0.7545** | **9.47 / 0.8093** |

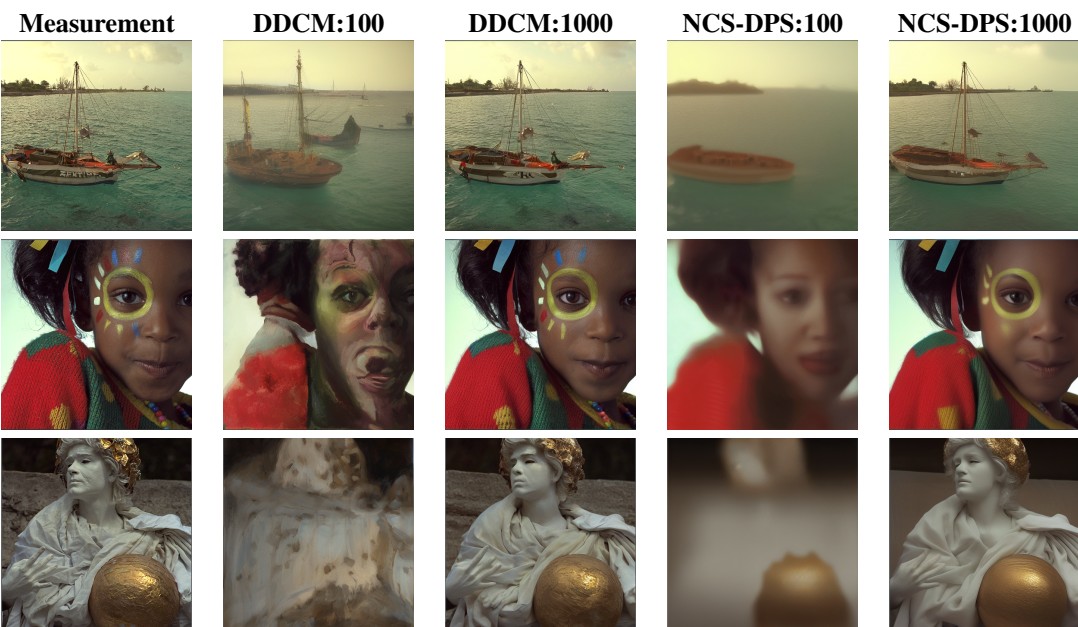

Figure 12: Compression results on Kodak24 images. Each row corresponds to a different image. Columns: Original, DDCM (100 steps), DDCM (1000 steps), NCS-DPS (100 steps), NCS-DPS (1000 steps).

