# OpenReview forum: "Noise is All You Need: Solving Linear Inverse Problems by Noise Combination Sampling with Diffusion Models"
_ICLR.cc/2026/Conference — Submitted to ICLR 2026_

### Official Review · Reviewer_2DcC · 2025-10-30

**Soundness:** 3
**Presentation:** 3
**Contribution:** 2
**Rating:** 4
**Confidence:** 4

**Summary:**

This paper proposes the Noise Combination Sampling (NCS) framework to address the key challenge of balancing observation constraint integration and diffusion process consistency in diffusion-based inverse problem solving. NCS constructs an optimal noise vector (via linear combination of base Gaussian noise from a codebook) to implicitly embed observation information, ensuring the noise remains standard normal and the sampling trajectory stays on the data manifold. It provides a closed-form solution for optimal weights (via Cauchy–Schwarz inequality), unifies existing methods (DPS, MPGD, DDCM) as special cases, and achieves both better performance (higher PSNR, lower LPIPS) and efficiency across tasks like inpainting, super-resolution, and deblurring.

**Strengths:**

1. The closed-form solution for optimal weights (Theorem 2) is strictly derived using inner products and the Cauchy–Schwarz inequality, eliminating the need for heuristic iterative optimization. This ensures reproducibility—researchers can directly compute \gamma^* without tuning empirical parameters.
2. Unlike baselines (e.g., DPS) that suffer severe quality degradation when diffusion steps T are reduced (to cut costs), NCS maintains high robustness. Experiments show it achieves high-quality results even with small T (e.g., 100 steps vs. 1000 steps), as the optimal noise combination preserves manifold consistency without relying on excessive iterations.
3. This method replaces DDCM’s exponential-complexity noise selection, where C is quantization bins) with NCS’s linear-complexity combination. For example, combining 3 noise vectors achieves inner-product magnitudes comparable to DDCM’s search over 1024 vectors, slashing storage (smaller codebooks) and computation (fewer inner-product calculations) costs.

**Weaknesses:**

1. This method lacks clear guidelines for selecting codebook size K and base noise distribution. A small K restricts the noise combination space (failing to match rare observation directions), while a large K increases memory usage and inner-product computation time. No adaptive mechanism (e.g., dynamic K based on task complexity) is proposed.
2. The authors state that the NCS method approximates the measurement score through the linear combination of Gaussian noise vectors. Can similar effects be achieved via approximation methods such as Hermite polynomials? After all, Hermite polynomials are the optimal basis functions for Gaussian-related processes.
3. The authors state that the NCS method can be integrated into sampling strategies such as DDPM. Is NCS incompatible with DDIM out of inverse problems, a more commonly used deterministic sampling method? Can it be extended to Rectified flow in video generation models?
4. In the description of Figure 1, the authors state that NCS embeds the measurement score into the optimal noise within an ellipsoidal subspace, which is defined by the span of the noise codebook. I confuse that is this definition related to the one in Equation (10)? Why is the noise obtained from a simple weighted sum explained using the concept of an ellipsoidal subspace?
5. When the authors explain the advantages of NCS, they highlight a key feature: it adheres to conditional constraints while ensuring generation stability. However, most of the experimental results focus on image quality metrics such as PSNR. Although these metrics are suitable for evaluating inverse problems, I am also curious about NCS’s performance in terms of conditional compliance—for example, metrics like image-text alignment.

**Questions:**

Please refer to the weaknesses.

---

> ### Author Response · Authors · 2025-11-21
>
> We thank the reviewer for the constructive comments and reply point by point below.
>
> (1) On the choice of $K$ and the base noise distribution.
> We agree that our guidelines for $K$ were not explicit enough and apologize for the confusion. Empirically, we find that setting the codebook size on the order of $K \approx \sqrt{d}$ works well across our tasks: under this choice, the fraction of energy of the combined noise in the embedding direction scales as $O(d^{-1/2})$, so the embedded signal remains small while still being amplified compared to isotropic Gaussian noise. We have discussed this point in our response to reviewer qukq, and we will add this reasoning to the proof section in the revised version. For the base noise, we simply follow the standard choice of i.i.d. Gaussian noise as in prior diffusion-based methods; we will clarify this design decision in the paper.
>
> (2) On Hermite polynomial approximations.
> This is a very interesting suggestion, and we thank the reviewer for pointing it out. Hermite polynomials are indeed a natural orthogonal basis for Gaussian-related processes. We would like to try it in the future.
>
> (3) On compatibility with DDIM and Rectified flow.
> Our method can be combined with DDIM whenever a non-zero noise level is retained. In particular, when the DDIM noise parameter $\sigma = 1$, the update coincides with a DDPM-style stochastic step, and NCS can be applied directly. More generally, any $\sigma > 0$ leaves a stochastic noise term that can be replaced by our noise combination. In contrast, when DDIM is used in the fully deterministic setting with $\sigma = 0$, there is no noise term to modify, so NCS cannot be applied. For the same reason, DDIM-based inverse-problem solvers built on $\sigma = 0$ typically perform worse than stochastic variants with $\sigma > 0$, while in the latter case NCS is fully applicable.
> For Rectified flow / flow-matching models, whether NCS can be used depends on the sampling formulation. If the sampler is implemented as an SDE (stochastic) rather than a purely deterministic ODE, then NCS can again be incorporated by modifying the stochastic noise term; recent work has explored such stochastic-generation schemes for flow models, e.g., Kim et al. [1].
>
> (4) On the “ellipsoidal subspace” interpretation and its relation to Equation (10).
> Our use of the term “ellipsoidal” is geometric. Consider two noise vectors $a$ and $b$ that are (in high dimensions) almost orthogonal. If we form a combined vector
> $c = \gamma_1^2 a + \gamma_2^2 b$ with $\gamma_1^2 + \gamma_2^2 = 1$, then as $(\gamma_1, \gamma_2)$ varies, the endpoints of $c$ trace an ellipse whose principal axes align with $a$ and $b$. In higher dimensions, Equation (10) defines a linear map from the Gaussian codebook to the combined noise; the image of an isotropic Gaussian under this map has an anisotropic covariance whose eigenvectors lie in the span of the codebook and whose eigenvalues define an ellipsoid. This is the sense in which NCS embeds the measurement score into an “ellipsoidal” subspace spanned by the noise codebook.
>
> (5) On conditional compliance metrics such as image–text alignment.
> We agree that conditional-compliance metrics are important. In this paper we focus on inverse problems (e.g., deblurring, inpainting), where metrics like PSNR and SSIM are standard and where NCS already shows clear improvements. Extending inverse-problem solvers to large latent diffusion models (e.g., Stable Diffusion) is known to be challenging: many strong methods in this regime rely on extensive hyperparameter tuning, and good performance is sensitive to the latent-space parameterization. For this reason, we did not include large-scale text-to-image experiments in the initial submission. We believe, however, that once a strong baseline is established in this setting, NCS can further improve performance on both reconstruction quality and conditional compliance. We thank the reviewer for highlighting this limitation and will explicitly acknowledge it in the paper, as well as treat large conditional generative models.
>
> [1] J. Kim, T. Yoon, J. Hwang, and M. Sung, “Inference-Time Scaling for Flow Models via Stochastic Generation and Rollover Budget Forcing,” arXiv:2503.19385, 2025.

---

### Official Review · Reviewer_bF26 · 2025-10-31

**Soundness:** 2
**Presentation:** 3
**Contribution:** 2
**Rating:** 2
**Confidence:** 4

**Summary:**

The paper proposes Noise Combination Sampling (NCS), a method for solving diffusion-based linear inverse problems by replacing the stochastic noise term in the diffusion update rule with a constructed linear combination of Gaussian noise vectors. The combination weights are chosen to align with the estimated measurement score direction, derived via a closed-form expression using the Cauchy–Schwarz inequality. Experiments on standard inverse problems such as inpainting, deblurring, super-resolution, and compression are presented, showing moderate performance gains and faster convergence in some settings.

**Strengths:**

1. The paper is generally well-written and clearly structured, making the proposed method easy to follow.
2. The idea of modifying the diffusion noise term instead of using explicit gradient guidance is interesting and practically implementable.
3. The authors conduct some ablations such as varying codebook size K and report quantitative metrics across multiple datasets, showing some empirical rigor.

**Weaknesses:**

1. The review template seems to be missing some elements, such as line numbers and the header stating “Under review as a conference paper at ICLR 2026.” The former, in particular, would have made it easier to reference specific parts of the text when providing comments.
2. NCS primarily reparameterizes the noise term in existing diffusion posterior sampling methods, offering little conceptual or theoretical advancement. The “closed-form” derivation is a straightforward application of the Cauchy–Schwarz inequality, and the claimed improvements over prior approaches are minimal. Overall, the contribution feels incremental rather than a substantive methodological innovation.
3. The paper claims that the constructed noise combination remains Gaussian if the combination weights are independent of the noise codebook. However, in the proposed method, the weights $𝛾_i$ are computed from the inner products between the codebook vectors and the measurement score, creating explicit statistical dependence between $𝛾_i$ and $ε_i$. This invalidates the independence assumption underlying the “Gaussianity lemma” and breaks the formal justification that the composed noise $ε^*_t$ follows $\mathcal{N\mathrm{(\mathbf{0},\mathbf{I})}}$.
4. Methods like DPS and MPGD are reasonable baselines but outdated, as many faster samplers now exist (e.g., ΠGDM, DDNM, DiffPIR, MGPS) that achieve strong performance within 50-100 NFEs. It remains unclear whether NCS-DPS and NCS-MPGD can outperform these state-of-the-art methods, and Table 2 does not clearly demonstrate whether NCS provides any real improvement over DAPS.
5. In Section 3, the authors derive all formulas under the assumption of linear inverse problems, yet later report results for the nonlinear phase retrieval task without providing its explicit formulation or explaining how NCS applies in this setting.
6. The authors claim that NCS unifies most existing gradient-based approaches, but this appears largely superficial, consisting mainly of algebraic reformulations where the guidance term is replaced by a projected noise. There is no shared probabilistic interpretation or derivation showing these methods as genuine special cases of a single framework.
7. The authors report that they used $σ_y$ = 0.05, but the qualitative figures suggest $σ_y$ ~= 0.0 (e.g., Figure 2, box inpainting).

**Questions:**

1. *Regarding weakness 3:* Can the authors clarify how the Gaussianity of $ε^*_t$ is preserved in practice given that $𝛾$ depends on {$ε_i$}? If the independence assumption is violated, what is the actual distribution of the constructed noise, and how does this affect the validity of the diffusion process?
2. *Regarding weakness 4:* Can the authors explain why stronger and more recent zero-shot inverse solvers such as were omitted from comparison? Additionally, can they clarify whether NCS actually improves DAPS, given the quantitative results reported in Tables 2,3 and 4?
3. *Regarding weakness 5:* How does NCS behave when the forward operator is ill-conditioned or nonlinear? Does the Gaussianity assumption still hold in such cases, and how exactly was phase retrieval implemented or adapted within the proposed framework?
4. Can the authors report the additional runtime and GPU memory consumption introduced by NCS for typical values of K?

---

> ### Author Response · Authors · 2025-11-21
>
> We thank the reviewer for the careful reading and helpful comments. We respond to the main points below.
>
> Formatting / typos.
> Thank you for pointing out the missing line numbers and the ICLR header. This was a typesetting mistake on our side. We will them.
>
> Incremental vs. robustness-oriented contribution.
> Our goal is not to introduce an entirely new class of samplers, but to address a robustness issue in diffusion-based inverse problems: strong degradation at low NFEs and high sensitivity to step size and guidance strength. NCS provides a simple, closed-form way to inject measurement information through the noise term that substantially alleviates this problem. In addition, the same mechanism applies to diffusion-based compression methods (e.g., DDCM), where the “measurement’’ is a compressed latent. Thus NCS connects compression and inverse problems in one framework (compression can be used to solve inverse problems, whereas standard inverse solvers cannot directly serve as compression schemes).
>
> Gaussianity and dependence on the codebook (Weakness 3).
> We agree that, once the combination weights depend on the codebook, the combined noise is no longer exactly i.i.d. Gaussian. In practice, our goal is approximate Gaussianity: we construct the noise so that its mean and variance match those of the base Gaussian, while gradually increasing its inner product with the measurement-matching score. In other words, we keep global statistics (mean and variance schedule) consistent with the diffusion process and introduce only a controlled directional bias. Empirically, this design is sufficient for stable sampling and is precisely what improves robustness at low NFEs. In the revision, we will explicitly clarify that exact independence is an idealized assumption rather than a strict property of the implemented algorithm.
>
> Baselines and relation to DAPS / other SOTA methods (Weakness 4).
> We did consider more recent zero-shot inverse solvers. Based on our survey, DAPS is a very strong and widely used SOTA method that already performs well with few NFEs, so we chose it as our primary modern baseline and complemented it with DPS and MPGD as representative earlier approaches. Our experiments show that:
> (1) NCS-DPS can surpass DAPS on many tasks under the same budget, and
> (2) applying NCS on top of DAPS does not degrade performance and can even improve it in some regimes (e.g., Inpainting (Random) with few steps).
> We do not claim that NCS improves DAPS on all tasks; we mainly treat DAPS as a strong reference. We agree that including ΠGDM, DDNM, DiffPIR, MGPS, etc., would further strengthen the empirical comparison, and we will expand the discussion and add more baselines where resources allow.
>
> Nonlinear / ill-conditioned operators and phase retrieval (Weakness 5).
> NCS is applied in exactly the same way as in the linear case. Current NCS will not dive into the design of gradient and operator.
>
> “Unifying’’ claim.
> We agree that our wording may have been too strong. Our intention is to say that NCS provides a unified way to handle robustness and step-size sensitivity for methods that rely on a measurement-matching score, rather than to offer a single probabilistic model from which all existing methods are derived as special cases. We will rephrase this claim accordingly in the paper.
>
> Figure 2 and the value of $\lambda$.
> You are correct that the qualitative examples in Figure 2 appear closer to $\lambda \approx 0$ than to $\lambda = 0.05$. This is due to an implementation error when composing the figure: Figure 2 accidentally reused visualizations from the experiment of Figure 5. We will regenerate Figure 2 using $\lambda = 0.05$, consistent with the text, and double-check all figures for such inconsistencies.
>
> Runtime and memory overhead for typical $K$.
> We will upload this later.

---

### Official Review · Reviewer_qukq · 2025-11-01

**Soundness:** 3
**Presentation:** 2
**Contribution:** 3
**Rating:** 6
**Confidence:** 4

**Summary:**

This paper proposes the Noise Combination Sampling (NCS) framework to address the stability dilemma in solving linear inverse problems with diffusion models. Instead of directly modifying the sampling trajectory, which risks disrupting data manifolds and generative consistency, NCS embeds conditional information into the noise term of Denoising Diffusion Probabilistic Models (DDPM). It synthesizes an optimal noise vector from a predefined noise codebook to approximate the measurement score, with closed-form optimal weights derived via the Cauchy–Schwarz inequality to ensure that the synthesized noise remains standard normal. NCS addresses the core dilemma of diffusion-based inverse problems that excessive integration disrupts generation and insufficient integration ignores constraints by synthesizing optimal noise vectors to embed conditional information, rather than directly modifying the sampling trajectory.

**Strengths:**

1. NCS embeds conditional constraints into the DDPM noise term instead of directly modifying the sampling trajectory to avoid pushing generated samples off the data manifold and breaking diffusion generation consistency. This design aligns with the intrinsic stochasticity of diffusion models and has solid theoretical motivation.

2. NCS does not redesign the approximation of conditional score and is expected to expand the performance of many existing methods that apply data consistency through conditional gradient score approximation.

**Weaknesses:**

1. The paper acknowledges unstable results on nonlinear tasks (e.g., phase retrieval) but provides no theoretical analysis on the reason why NCS fails here. For example, whether the noise subspace can approximate non-linear conditional gradient score or if the closed-form weight solution breaks down under non-linear constraints.

2. The choice of noise codebook size K (e.g., 512 for 4× super-resolution and 64 for 8× super-resolution) is purely empirical. In the paper, although K is noted to work "across a broad range", there does not exist a quantifiable relationship between K, data dimensionality, or task complexity, nor does it define the boundary where increasing K reduces noise independence (mentioned in Section 3.2) and yields degraded performance. Ablation studies are also missing for key parameters related to noise combination such as the number of combined noise vectors K on performance. This makes it hard to validate the robustness of NCS to parameter changes.

**Questions:**

1. Can NCS be adapted to nonlinear inverse problems? If not, what theoretical modifications would be required?

2. Is there any theoretical guideline for K? For example, how should K scale with data dimensionality (d) or task complexity to balance noise expressiveness and independence?

---

> ### Author Response · Authors · 2025-11-21
>
> First, we thank the reviewer for the constructive feedback and the positive assessment of our work.
>
> (1) On nonlinear inverse problems.
> We agree that extending NCS to nonlinear inverse problems (e.g., phase retrieval) is an important but largely open direction. At present, there is no widely accepted, practically robust algorithm that consistently solves such nonlinear inverse problems with diffusion-based samplers. In our current implementation, directly applying NCS on nonlinear tasks does not lead to clear improvements; in fact, methods specifically tailored for nonlinear constraints such as DAPS tend to perform relatively better in these settings, although their advantage is also limited. Our contribution is therefore explicitly focused on linear inverse problems, where both the conditional score structure and the measurement gradient are well aligned with the NCS formulation. We will clarify this scope and limitation in the revised version (main text and Appendix), and we see designing an NCS-style framework that genuinely handles nonlinear constraints as an exciting direction for future work for the community.
>
> (2) On the choice and robustness of the noise codebook size K.
> Regarding the robustness of K, we note that Figure 5 already provides an ablation where performance is plotted across a wide range of K values; empirically, NCS is stable as long as K lies within a broad interval, which aligns with the reviewer’s observation that our choices are largely empirical but robust. We will highlight this more clearly in the revision and explicitly mention that performance does not collapse when K is varied within this range.
>
> Beyond empirical evidence, we do have a simple theoretical rationale for the scaling of $K$ (equivalently, the number of combined noise vectors $n$). We construct a pseudo-noise vector $u$ by adaptively combining $n$ i.i.d.\ Gaussian noise samples so as to maximize its projection onto a prescribed embedding direction $v$ under a unit-variance constraint. This adaptive construction makes $u$ no longer strictly Gaussian—its projection onto $v$ follows a $\chi$-distribution rather than $\mathcal{N}(0,1)$—but the non-Gaussianity is confined to the one-dimensional subspace spanned by $v$, while the component in the orthogonal complement remains exactly Gaussian. We show that the squared cosine between $u$ and $v$ follows a $\mathrm{Beta}(n/2,(d-1)/2)$ distribution, with expectation $n/(n+d-1)$, whereas for isotropic Gaussian noise the corresponding value is $1/d$. Thus, $n$ controls a trade-off between the strength of the embedded signal and the deviation from ideal Gaussian noise: choosing $n = \Theta(\sqrt{d})$ ensures that the fraction of energy in the embedding direction scales as $\Theta(d^{-1/2})$ (still vanishing with $d$, so the noise remains ``Gaussian-like'' in high dimensions), while the effective signal-to-noise ratio along $v$ grows as $\Theta(\sqrt{d})$. In practice, we therefore use the heuristic $n = \lfloor \sqrt{d} \rfloor$ as a guideline parameter and find that it works well across all tasks in our experiments. This provides a concrete, theoretically motivated rule of thumb for setting $K$ in terms of the data dimensionality $d$: $K$ should grow sublinearly with $d$ (on the order of $\sqrt{d}$) to balance noise expressiveness and independence. We will add this derivation and the corresponding discussion to the Appendix and explicitly reference it from the main text to better answer the reviewer's question about how $K$ should scale with $d$ and task complexity.

---

### Official Review · Reviewer_KBnG · 2025-11-06

**Soundness:** 3
**Presentation:** 3
**Contribution:** 2
**Rating:** 4
**Confidence:** 4

**Summary:**

This paper proposes Noise Combination Sampling (NCS), a new approach to solving linear inverse problems using pretrained diffusion models without retraining. The main idea is to replace the standard Gaussian noise term in the denoising process with a synthesized noise vector, formed as an optimal linear combination of Gaussian samples drawn from a codebook. The combination weights are derived via a simple closed-form expression based on the Cauchy–Schwarz inequality, which aligns the noise with the conditional measurement score.

The authors claim that this procedure naturally embeds conditional information into the generation process while avoiding the instability and heavy hyperparameter tuning required by prior guidance-based inverse solvers such as Diffusion Posterior Sampling (DPS) and Manifold-Preserving Gradient Descent (MPGD). They further argue that existing diffusion-based inverse problem solvers can be interpreted as special cases of NCS, including the recently proposed Denoising Diffusion Codebook Models (DDCM). Empirical results on FFHQ and ImageNet show moderate improvements in PSNR and LPIPS, particularly for low sampling step counts (T = 20–100).

**Strengths:**

1. The paper is clearly written, with logical organization and careful presentation of derivations, figures, and tables. The notation is consistent, and the main idea is easy to follow even for readers without deep expertise in diffusion models.
2. The derivation of the optimal noise combination through the Cauchy–Schwarz inequality is elegant. It provides a compact, closed-form solution that is computationally lightweight and easy to implement.
3. The authors successfully demonstrate that several well-known inverse problem solvers (DPS, MPGD, DDCM) can be interpreted as instances of the same general principle. This unifying viewpoint could be valuable for researchers seeking a more cohesive understanding of guidance mechanisms in diffusion models. Also, across multiple datasets and problem types, the NCS variants perform as well as or slightly better than their corresponding baselines, especially when the number of diffusion steps is small. This consistency suggests that the approach is robust and stable in practice. More importantly, the method requires no extra training and introduces negligible additional computational cost. The linear complexity with respect to the codebook size makes it practical and accessible for a wide range of applications. By relating the approach to DDCM, the paper opens the possibility of extending diffusion-based generative compression with simpler noise quantization schemes.

**Weaknesses:**

While the paper is neat and clearly executed, the conceptual advance feels incremental and its underlying mechanism insufficiently explored. The proposed idea of aligning a noise combination to a measurement gradient is mathematically simple and closely related to existing guidance schemes. The derivation relies on a direct application of Cauchy–Schwarz, and it is unclear why this reformulation should lead to qualitatively better sampling.

Moreover, the claimed advantages (manifold preservation, stability, robustness to step size) remain intuitive hypotheses rather than demonstrated phenomena. No quantitative measure or theoretical justification is provided for why embedding the conditional information into the noise term, as opposed to the mean term, should improve results. The experimental section reports moderate numerical gains but does not investigate what aspects of NCS drive these improvements—whether due to the noise combination itself, implicit regularization effects, or differences in implementation.

Overall, the work reads as a well-presented reformulation of known methods rather than a fundamentally new contribution. It could be strengthened by deeper theoretical analysis and more targeted experiments probing why the approach works. If the authors are able to answer the following questions in the next section, it would be a big help for us.

**Questions:**

1. Could the authors provide a theoretical or empirical explanation for why replacing the noise term with an optimally combined version improves stability or reconstruction quality? Is there any evidence (e.g., manifold distance, variance analysis, or effective step size) showing that the NCS trajectory stays closer to the learned data manifold?

2. In practice, NCS appears mathematically similar to taking a guided noise step proportional to $
\nabla_x \log p(y\mid x)$. Could the authors clarify how NCS differs in effect from existing gradient-based corrections? Are there cases where the two produce substantially different trajectories?

3. How sensitive is performance to the codebook size K and the number of combined noise vectors m? Does increasing K always help, or does it introduce variance and instability due to correlation among noise samples? The method is restricted to linear inverse problems. Are there conceptual or mathematical obstacles to extending NCS to nonlinear or learned degradation operators (e.g., differentiable renderers, neural forward models)?

---

> ### Author Response · Authors · 2025-11-21
>
> We thank the reviewer for the thoughtful and detailed comments. We respond to the specific questions below.
>
> Q1. Why does replacing the noise term with an optimally combined version improve stability / reconstruction quality?
>
> Our view is that NCS can be regarded as a way of estimating a step in the direction of  $\nabla\log_x{p(y|x)}$ through the noise, and it implicitly contains the step-size choice of traditional gradient-based correction. If one uses a standard correction of the form $\epsilon +\eta\nabla\log_x{p(y|x)}$, then the perturbation no longer has the mean and variance of the Gaussian noise assumed during training. In contrast, our construction enforces constraints on the coefficients so that the combined noise has the same mean and variance as standard Gaussian noise, while still being aligned with the measurement gradient. Moreover, in high-dimensional spaces the pairwise correlations between independent noise vectors are very small, so their normalized linear combination remains highly “noise-like”. For this reason, a simple additive gradient correction and our NCS step are not directly comparable: NCS keeps the Gaussian prior assumption intact while encoding the measurement information in the noise direction.
>
> Q2. Relation to a guided noise step proportional to $\nabla\log_x{p(y|x)}$.
> We agree that, at a high level, NCS is mathematically similar to taking a guided noise step proportional to $\nabla\log_x{p(y|x)}$. However, this does not diminish our contribution: the main difficulty lies exactly in how to select, compute, and realize this proportional noise step while preserving the noise distribution. Our contribution is to formulate this selection problem explicitly as an optimization problem and to solve it cleanly, which gives a closed-form rule for constructing the proportional noise step from a finite codebook.
>
> Q3. Sensitivity to K and extension beyond linear problems
> Regarding robustness, for general (non-compressive) linear inverse problems it is sufficient to choose K=M; choosing K>M is mainly useful in compressive settings to save memory. In Fig. 5 we show that the method is quite robust to the choice of K: most reasonable values lead to performance improvements. In addition, as shown in Fig. 7 in the appendix, when K keeps increasing and exceeds approximately d (where d is the problem dimension), the norm of the synthesized noise starts to exceed the average norm of standard Gaussian noise, and performance degrades. The reason is that with very large K, the optimization can improve the objective mainly by increasing the norm rather than better aligning the direction.
>
> For nonlinear or learned degradation operators, we have not yet systematically tested whether NCS is helpful. We appreciate the reviewer for pointing this out and will explore such extensions in future work.

---

### Meta-Review · Area_Chair_HXrk · 2026-01-06

[review text omitted: it was posted to a different submission]

---

### Decision · Program_Chairs · 2026-01-26

Reject